# Adaptive control of movement deceleration during saccades

**Simon P. Orozco**⊙*, **Scott T. Albert**⊙, **Reza Shadmehr**⊙*

Laboratory for Computational Motor Control, Dept. of Biomedical Engineering, Johns Hopkins School of Medicine, Baltimore, Maryland, United States of America

\* sorozco2@jhu.edu(SPO); shadmehr@jhu.edu (RS)

## Abstract

As you read this text, your eyes make saccades that guide your fovea from one word to the next. Accuracy of these movements require the brain to monitor and learn from visual errors. A current model suggests that learning is supported by two different adaptive processes, one fast (high error sensitivity, low retention), and the other slow (low error sensitivity, high retention). Here, we searched for signatures of these hypothesized processes and found that following experience of a visual error, there was an adaptive change in the motor commands of the subsequent saccade. Surprisingly, this adaptation was not uniformly expressed throughout the movement. Rather, after experience of a single error, the adaptive response in the subsequent trial was limited to the deceleration period. After repeated exposure to the same error, the acceleration period commands also adapted, and exhibited resistance to forgetting during set-breaks. In contrast, the deceleration period commands adapted more rapidly, but suffered from poor retention during these same breaks. State-space models suggested that acceleration and deceleration periods were supported by a shared adaptive state which re-aimed the saccade, as well as two separate processes which resembled a two-state model: one that learned slowly and contributed primarily via acceleration period commands, and another that learned rapidly but contributed primarily via deceleration period commands.

**Data Availability Statement:** Data is available on https://osf.io/k35ts/.

**Funding:** The work was supported by grants from the following organizations to R.S.: National Science Foundation (CNS-1714623) National

## Author summary

A theoretical model posits that motor learning is supported by two independent processes, one that learns robustly from error but has poor retention, and another that learns more slowly but has strong retention. Here, we observed the potential signatures of these adaptive processes during control of saccades. The results suggest that distinct adaptive controllers may contribute to the acceleration and deceleration phases of a single movement.

## Introduction

As you read this text, your eyes must accurately move from word to word, and from the end of one line to the start of the next. This accuracy is partly due to the cerebellum [1,2], which

Institutes of Health (R01-NS078311, R01-NS096083) Office of Naval Research (N00014-15-1-2312). S.O. was supported by a fellowship from the National Institutes of Health (1F31-NS108731). The funders had no role in study design, data collection and analysis, decision to publish, or preparation of the manuscript.

**Competing interests:** The authors have declared that no competing interests exist.

monitors errors that are made, learns from them, and corrects the motor commands that guide the subsequent saccade [3]. However, this process of adaptation exhibits a remarkable pattern: when the errors induce adaptation, saccade endpoints gradually change, and when the errors reverse direction, inducing extinction, endpoints return to baseline. Yet, following passage of time, and without presence of errors, the saccade endpoints revert back toward the previously adapted state [4,5]. This "spontaneous recovery" is a fundamental feature of many learning paradigms, including classical conditioning [6], fear conditioning [7], and various forms of motor and perceptual learning [4,8–14]. However, its origin remains a mystery.

A mathematical model [8,15] suggests that during learning, changes in behavior may be supported by two adaptive processes: a fast adaptive process that has high sensitivity to error but suffers from poor retention, and a slow adaptive process that has poor sensitivity to error but benefits from robust retention. In this model, learning followed by extinction produces competition between the fast and the slow processes. Following extinction training, with passage of time the fast process decays, inducing spontaneous recovery of the previously learned behavior. Here, we searched for signatures of these hypothesized processes in saccadic eye movements.

We began with the observation that if the brain experiences a visual error following completion of a saccade (i.e., target is not on the fovea), it learns from that error, producing a change in the motor commands that guide the subsequent saccade [5]. Importantly, the early and the late components of the eye's trajectory exhibit different patterns of change [16]: whereas the early part of the trajectory adapts little, the late part of the same saccade adapts more. Moreover, with passage of time during set breaks, the early part of the trajectory shows little or no forgetting, but the late part of the same trajectory suffers from more forgetting [16]. Could it be that the slow and fast timescales of adaptation are expressed within different parts of a single movement?

To investigate this question, we induced adaptation by presenting errors that were perpendicular to the primary saccade. This made the adapted commands orthogonal to the original commands, thus allowing for isolation of the learned behavior. As the brain adapted, the response during the acceleration and deceleration phases differed from one another: the acceleration period commands adapted slowly and exhibited robust retention, whereas the deceleration period commands adapted rapidly and exhibited poor retention. Thus, commands during the early and late parts of a single saccade appeared to express certain properties of the hypothesized slow and fast processes.

Next, we performed an experiment in which the perturbations were random. When we isolated the adaptive response to a single error, we made a surprising discovery: following experience of error, the correction to the saccade on the subsequent trial was expressed only during deceleration.

Together, these results suggest that the commands that guide a single saccade may be supported by one adaptive controller that largely contributes to the acceleration period, and another that contributes more to the deceleration period. In principle, spontaneous recovery could arise from the differing error sensitivity and retention properties of these controllers. But when we tested this idea, a third adaptive state emerged, one that appeared to re-aim the saccade's trajectory and was shared across the acceleration and deceleration periods. Spontaneous recovery occurred when this shared state interacted with the slow and fast states expressed during the acceleration and deceleration periods.

## Results

We asked healthy human participants (n = 101) to make a saccade toward a visual target at 15˚. As the primary saccade started, we moved the target perpendicular to the primary movement by 5˚, thus presenting a visual error that encouraged adaptation.

## Deceleration period commands adapted with high sensitivity to error, but exhibited poor retention

In Exp. 1, participants (n = 24) made primary saccades in the vertical direction (80% of the trials, Fig 1A). To induce adaptation, after presentation of the target we jumped it horizontally at saccade onset so that the movement ended with an error perpendicular to its primary direction. This error induced adaptation, resulting in a change in the motor commands that guided subsequent primary saccades. We quantified the change in motor commands via the saccade's horizontal velocity (S1A Fig), which grew during perturbation trials but decayed during error clamp trials (S1D Fig) and following set breaks (30 sec in duration, S1B Fig). To analyze these patterns of adaptation, we focused on the time course of each saccade's horizontal velocity.

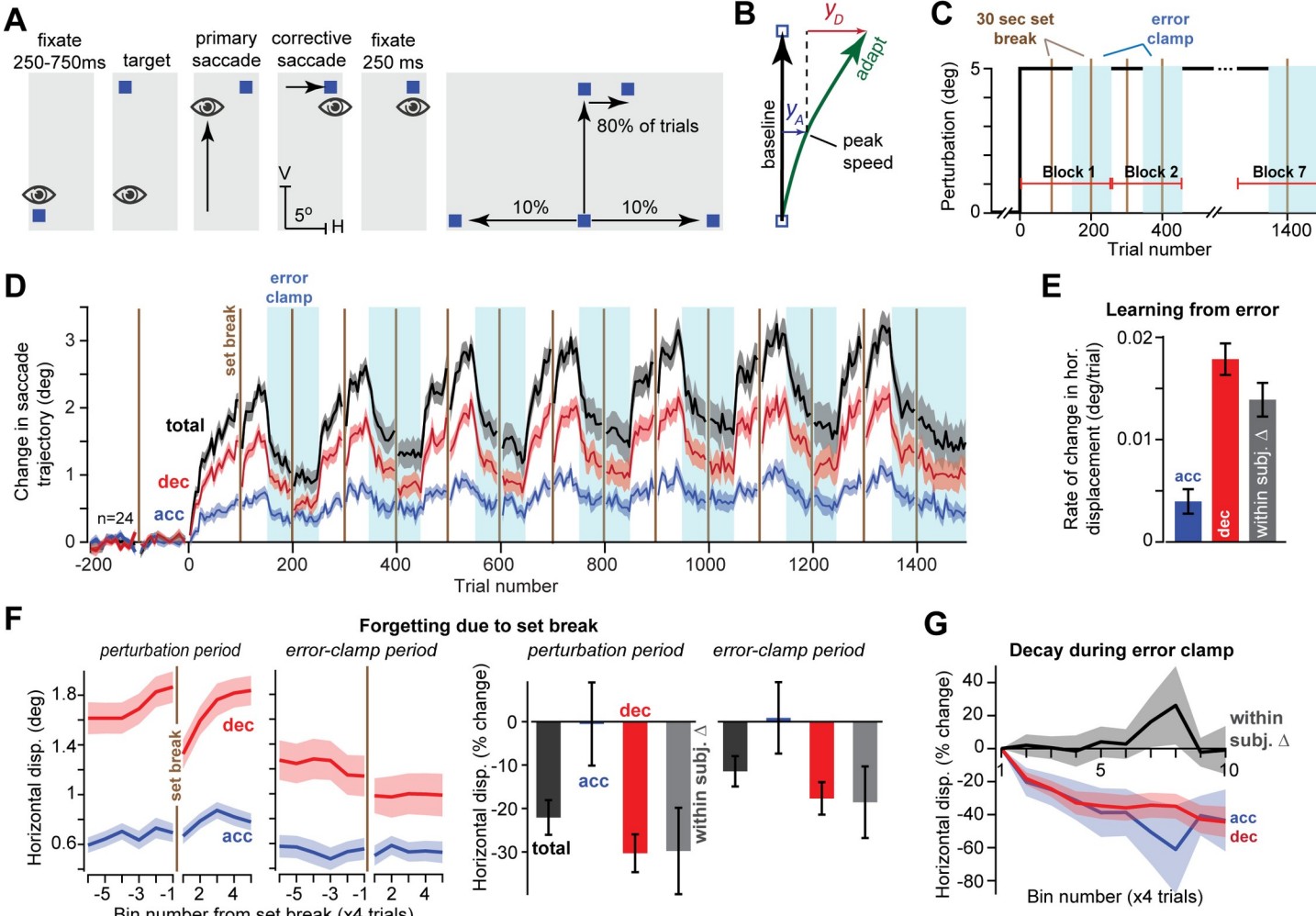

**Fig 1. Acceleration and deceleration period commands exhibit some of the properties of the fast and slow system. A**. Experiment 1 trial structure. Following a vertical primary saccade, subjects experienced a rightward horizontal endpoint error. **B**. To quantify adaptation, we measured the horizontal displacement of the adapted saccade with respect to baseline saccades. We integrated horizontal eye velocity over the acceleration and deceleration periods to determine the displacement during each period. **C**. Training blocks. Each block consisted of four periods: initial adaptation, set break (brown line), re-adaptation, error clamp. **D**. Horizontal displacement during the acceleration period was smaller than during the deceleration period, but exhibited little or no forgetting at set breaks. **E**. Trial-to-trial rate of change in the horizontal displacement produced by the acceleration and deceleration period commands during the perturbation blocks. Rate of change was faster in the deceleration period. **F**. Effect of set break on the horizontal displacement in the perturbation blocks and error clamp blocks. At left we show adaptation aligned to set breaks and averaged across all periods. Bin size is 4 trials. Decay was much greater in deceleration period commands. At right, we calculated the percent loss during set breaks in the perturbation and error clamp periods (depicted at left). **G**. Decay in the acceleration and deceleration period commands during the error clamp period. Error bars are between subject SEM. Bin size is 4 trials.

Our prior work (Chen-Harris et al. 2008) suggested that adaptation produced changes in saccade trajectory that differed between the early and late components of a single movement; the commands that arrived near the end of a saccade adapted nearly twice as much as the commands that initiated the saccade (Fig 1B). This suggested that peak speed may be a useful marker to divide a saccade into its early and late periods. Thus, we divided each saccade based on the timing of its peak speed, which remained highly consistent through the entire experiment (S1E Fig).

We used the timing of the peak speed to divide each saccade into an acceleration phase and a deceleration phase. Peak speed was defined as $\sqrt{\dot{H}^2 + \dot{V}^2}$, where $\dot{H}$ is horizontal velocity and $\dot{V}$ is vertical velocity. To determine how acceleration and deceleration period commands contributed to the learned response, we integrated the horizontal velocity during each phase of the saccade (Fig 1B), yielding the displacement during the acceleration and deceleration periods (Fig 1D). The displacement differed across the acceleration and deceleration periods in several ways.

First, deceleration period commands made larger contributions to adaptation than acceleration period commands (Fig 1D). Indeed, in response to error, deceleration period commands adapted at a faster rate (Fig 1E) than the acceleration period commands (RMANOVA, trial bin by movement phase interaction, $F(4,92) = 13.02$, $p = 1.93X10^{-8}$, within subject difference of $0.019 \pm 0.003$ deg/trial, $t(23) = 6.92$, $p = 4.72X10^{-7}$).

Second, although deceleration period commands adapted more rapidly, they suffered a substantial loss after a set break (Fig 1F, left). To quantify this loss, we calculated the percent change in displacement with respect to before set break onset (Fig 1F, right), for each participant that achieved at least 0.25° of adaptation (see Methods) in each movement phase prior to the set break. Indeed, set breaks caused large decay in deceleration period commands. For example, when set breaks occurred during the perturbation trials, we observed $30 \pm 4\%$ decay (Fig 1F, right, perturbation period, $t(21) = -6.95$, $p = 7.24X10^{-7}$). Set breaks during the error clamp trials also produced significant decay (Fig 1F, right, error clamp period, $-18 \pm 4\%$, $t(19) = -4.73$, $p = 0.0001$), but the decay was larger during the perturbation periods (within subject difference of $7 \pm 3\%$, $t(20) = 2.38$, $p = 0.027$).

In contrast to the deceleration period commands, the acceleration period commands showed a strong resistance to set breaks. During both perturbation periods and error clamp periods, set breaks led to much greater loss in deceleration commands than acceleration commands (Fig 1F, right; perturbation period, within subject difference: $-29.8 \pm 10\%$, $t(21) = -2.99$, $p = 0.007$; error clamp period, within subject difference: $-19 \pm 8\%$, $t(19) = -2.25$, $p = 0.037$). In fact, we detected no significant decay in acceleration period commands in either the perturbation periods (Fig 1F, perturbation period, $-0.5 \pm 9.6\%$, $t(21) = -0.05$, $p = 0.96$) or the error clamp periods (Fig 1F, error clamp period, $1 \pm 8\%$, $t(19) = 0.11$, $p = 0.92$). The stability in the acceleration period commands was also observed when decay was operationalized as the total change in adaptation (in deg) without normalization to the levels before the set break (S2B Fig).

We were concerned that our inability to detect set break decay in the acceleration period commands may have been because of their small magnitude. Thus, we searched for trials where deceleration period adaptation was as small in magnitude as the acceleration period commands and observed that once again, decay was present in the deceleration period commands but not acceleration (S2C Fig).

Surprisingly, while acceleration periods commands exhibited resistance to decay at set breaks, they exhibited substantial decay during the error clamp blocks (Fig 1G). Indeed, both the acceleration and deceleration period commands exhibited substantial decay during the

error clamp period. In fact, their decay rates did not differ (RMANOVA, main effect of movement phase, $F_{(1,23)} = 0.69$, $p = 0.42$, trial by movement phase interaction $F_{(39,897)} = 1.27$, $p = 0.13$; rates $-0.68 \pm 0.31\%$/trial vs. $-1.17 \pm 0.21\%$/trial; test of within subject difference $t_{(23)} = 1.53$, $p = 0.14$).

To summarize, the brain learned to alter vertical saccades by generating horizontal commands. However, these horizontal commands were not uniformly expressed during the saccade. Rather, the deceleration period commands adapted rapidly but exhibited poor retention during set breaks. In contrast, the acceleration period commands adapted slowly but exhibited little or no decay during set breaks. These observations were reminiscent of a two-state model of adaptation, with deceleration commands resembling a "fast state" and acceleration commands resembling a "slow state". However, there was one aspect of behavior that did not agree with the model: during the error clamp trials, both the acceleration and deceleration period commands decayed at similar rates.

## Control trials: deceleration period commands were specific to adapted saccades

At conclusion of training (Fig 1D), in response to a vertical target at (0˚,15˚) (horizontal and vertical position), the brain generated commands that moved the eyes to (3˚,15˚); the horizontal commands steered the eyes by roughly 3˚ as the eyes traveled 15˚ along the vertical direction. This 3˚ change in horizontal position of the eyes was largely due to commands that arrived during the deceleration period. With that said, the deceleration period lasted approximately 50 ms, twice as long as the period devoted to acceleration, which lasted approximately 20 ms (S1 Fig). Could it be that the acceleration period's smaller contribution to the adaptive response was simply due to its shorter duration?

To consider this possibility, we compared the trajectory of adapted saccades to that of oblique saccades which had the same endpoint and similar acceleration ($20.71 \pm 1.02$ms vs $20.04 \pm 0.71$ms) and deceleration ($54.00 \pm 1.56$ms vs $49.35 \pm 1.05$ms) periods, but were made in a control condition in which the primary target was presented at (3˚,15˚) as shown in Fig 2A, top-right.

In the control condition (baseline period prior to adaptation), we occasionally presented the target at an oblique location: (1˚,15˚), (2˚,15˚), . . ., or (5˚,15˚). Fig 2A plots the horizontal velocity in the control and adapted conditions, matched across different horizontal displacements. The saccades in the control and adapted conditions differed markedly. The top plot of Fig 2A displays horizontal velocities for saccades that ended with a 3˚ horizontal displacement. In the control condition, the acceleration and deceleration period commands contributed equally to the horizontal displacement. In contrast, in the adapted state, the 3˚ endpoint displacement was achieved mainly via commands that arrived in the deceleration period (within-subject 2-way ANOVA, main effect of adaptation vs. control $F_{(1,23)} = 5.80$, $p = 0.02$, main effect of acceleration vs. deceleration $F_{(1,23)} = 1.33$, $p = 0.26$, interaction $F_{(1,23)} = 49.55$, $p = 3.58 \times 10^{-7}$). This pattern repeated across other saccade magnitudes (Fig 2B, within subject difference between percent contribution of acceleration and deceleration commands collapsed across 1, 2, and 3 deg endpoints, baseline vs. adapt, $t_{(23)} = 5.13$, $p = 3.42 \times 10^{-5}$).

Therefore, in both control and adapted saccades, the deceleration period lasted roughly twice the length of the acceleration period. However, during control saccades, the endpoint horizontal displacement was due to equal contributions from acceleration and deceleration period commands (Fig 2B, gray bars, deceleration minus acceleration, $-10 \pm 11\%$, $t_{(23)} = -0.91$, $p = 0.37$). In contrast, during adapted saccades, a majority of the endpoint displacement was due to the commands that arrived during the deceleration period (blue bars in Fig 2B, deceleration minus acceleration: $34 \pm 5\%$, $t_{(23)} = 7.12$, $p = 2.98 \times 10^{-7}$). Thus, even though the

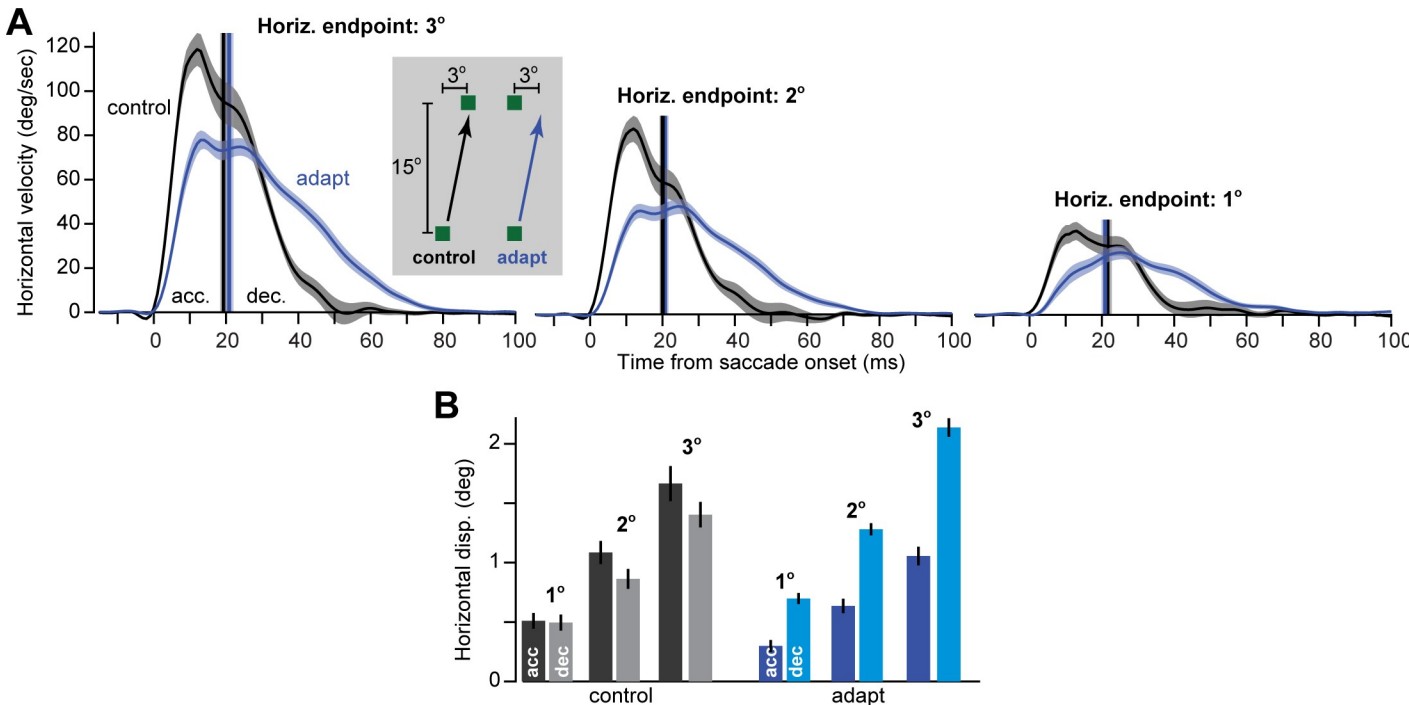

**Fig 2. In Experiment 1, adapted saccades arrive at an endpoint with a trajectory that is quite different from control saccades that arrive at the same endpoint. A.** Horizontal velocity of control saccades and adapted saccades. In control trials, targets are presented at (+1°,15°), (+2°,15°), and (+3°,15°). In adapted saccades, the same endpoint is achieved but through learning. Adapted saccades exhibit a smaller horizontal displacement during the acceleration period, but a larger displacement during the deceleration period (with respect to baseline). **B.** Control saccade have roughly equal horizontal displacement for the acceleration and deceleration period commands. In comparison, an adapted saccade has most of its horizontal displacement due to the deceleration period commands. Error bars are between subject SEM.

oculomotor system could have accomplished the desired horizontal displacement using equal contributions from acceleration and deceleration, as it did in the control condition, during adaptation it was somehow limited to expressing its contributions mainly during deceleration.

## Both the acceleration and deceleration period commands exhibited spontaneous recovery

Experiment 1 suggested that acceleration and deceleration commands resembled the slow and fast states of adaptation. In the framework of the two-state model, during extinction the fast state must compete with the slow state, thus resulting in spontaneous recovery [8]. We tested this possibility in Experiment 2.

Subjects (n = 40) made primary saccades to a target at 15°, and then experienced a +5° error perpendicular to the direction of the target (Fig 3A). We then reversed the perturbation direction (-5°) to encourage extinction of the acquired memory. After this extinction phase, a block of error clamp trials followed to assess spontaneous recovery. In separate groups of subjects, we tested adaptation to horizontal and vertical errors (Fig 3A). While horizontal errors induced greater learning, behavior was qualitatively similar across groups (S3 Fig), and thus the two groups were combined in Fig 3.

At the end of the initial +5° perturbation period, saccade endpoint had changed by roughly 3° (trials labeled $t_1$, Fig 3B, black). Reversing the direction of the visual error encouraged extinction, and by the end of the extinction trials endpoint displacement was less than 0.5° (trials labeled $t_2$, Fig 3B). In the error clamp period that followed, the saccades exhibited

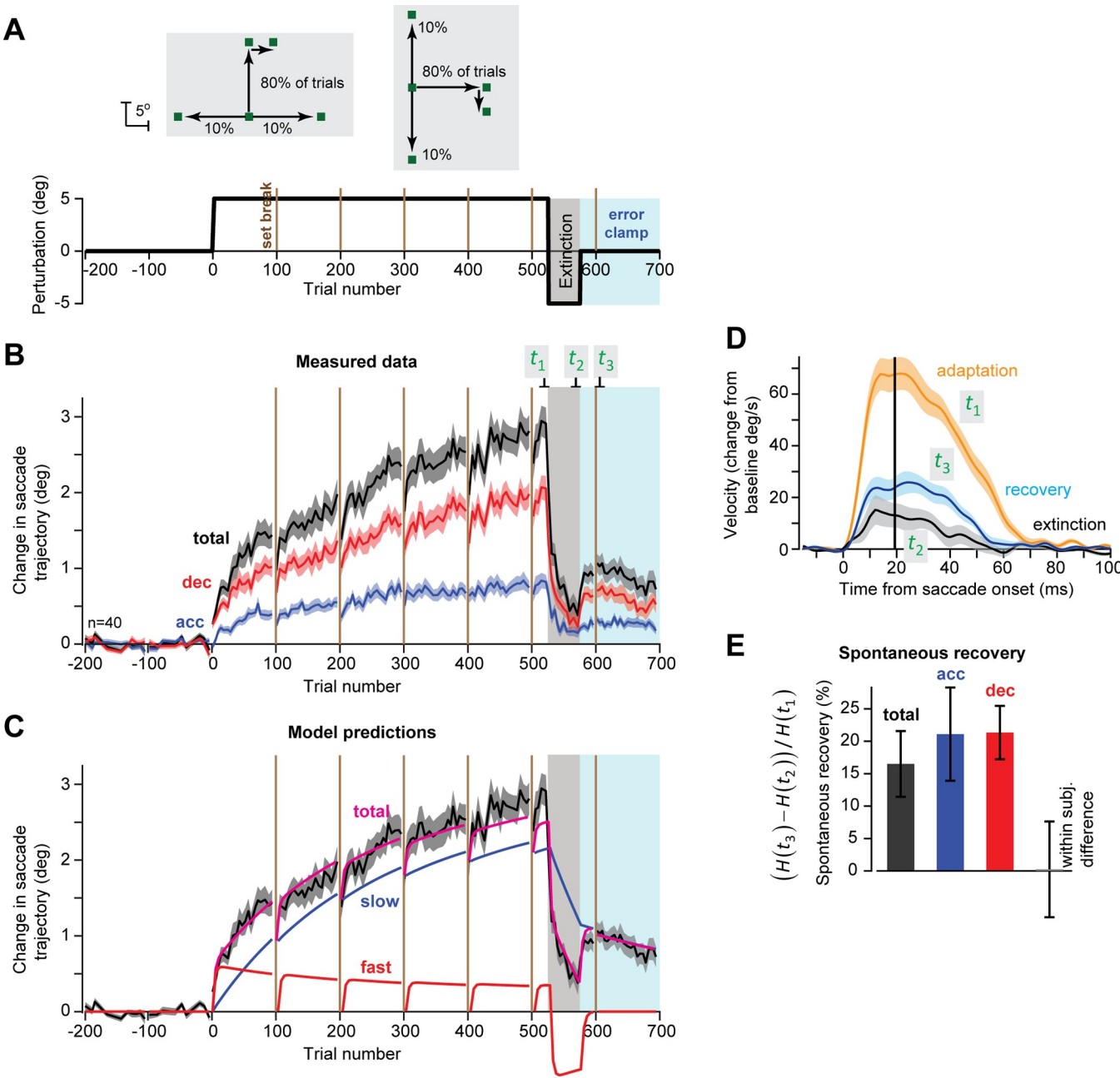

**Fig 3. Experiment 2.** Both acceleration and deceleration period commands exhibit spontaneous recovery. **A**. Experimental design. Perturbation block was followed by an extinction block, and then a block of error clamp trials. Half of the subjects experienced horizontal errors following vertical saccades (left panel), while the other half experienced vertical errors following horizontal saccades (right panel). **B**. Measured data. **C**. A standard two-state model was fit to the measured data (trace labeled total). The fast and slow states that are predicted by the model are shown. **D**. Eye velocity in the direction of error during the trials at end of adaptation ($t_1$), at end of extinction ($t_2$), and during error clamp ($t_3$). **E**. Spontaneous recovery, defined as [$H(t_3)−H(t_2)$]/$H(t_1)$.

spontaneous recovery: endpoint displacement with respect to the target increased from 0.5° to around 1° (from $t_2$ to $t_3$, Fig 3B and 3D). We quantified spontaneous recovery as the change in displacement from end of extinction trials (time $t_2$) to the error clamp block (time $t_3$) scaled by the level of adaptation at $t_1$: [$H(t_3)−H(t_2)$]/$H(t_1)$. Saccade endpoint exhibited roughly 16.5% spontaneous recovery ("total", Fig 3E, t(34) = 3.26, p = 0.002).

We next considered how well the standard two-state model could account for the measured data. We fit the model to the total displacement that we had measured on each trial (Fig 3C, magenta) and asked whether the fast and slow states of the model could match the acceleration and deceleration period commands. The observed data differed markedly from the model's states. Whereas the model predicted that performance in the late stages of training would be dominated by the slow state, the observed behavior was in fact dominated by the deceleration period, a putative fast state. More importantly, while the model predicted that spontaneous recovery was due to differential decay in the slow and fast states, the data showed something entirely different: both acceleration and deceleration periods exhibited spontaneous recovery (Fig 3D and 3E; 21±7%, t(34) = 2.93, p = 0.006 and 21±4%, t(34) = 5.19, p = 9.87X10$^{-6}$ respectively). Indeed, spontaneous recovery did not differ for the acceleration and deceleration periods (Fig 3E; within subject difference, 0.24±7%, t(34) = 0.03, p = 0.97).

In summary, when extinction training followed adaptation, in the subsequent error clamp trials there was spontaneous recovery of saccade endpoints toward the initially learned behavior. However, this was not caused by differential decay patterns in the acceleration and deceleration commands. Rather, the commands during both periods exhibited spontaneous recovery. Thus, despite the fact that acceleration period commands adapted slowly and exhibited resistance to decay, and deceleration period commands adapted rapidly but were susceptible to decay, their dynamics during extinction and spontaneous recovery were inconsistent with the standard 2-state model.

## Alternative models of spontaneous recovery

To better understand the results, we considered two possibilities. In Model 1 (Fig 4A), we imagined a single adaptive controller with multiple timescales of learning. In this scenario, a fraction of the learning was expressed during acceleration, while the remainder was expressed during deceleration. This model is akin to the one suggested in Chen-Harris et al. [16]: a forward model monitors the ongoing motor commands and corrects the movement as it takes place. Adaptation results in changes in both the acceleration and deceleration phases, but because the deceleration phase is longer, a greater fraction is expressed later.

A critical component of Model 1 was that learning and forgetting were coupled in the two parts of the movement. For example, if there was partial forgetting that affected the deceleration period, it would also affect the motor commands during the acceleration period. Because this aspect of the model was inconsistent with our data, we considered a variant. In Model 2 (Fig 4B), we imagined two controllers that each had a single timescale of learning. One controller learned to aim the saccade toward a new location, and another learned to independently modulate commands during the acceleration and deceleration periods.

We used state-space equations to represent Models 1 and 2 (see Methods). These models were parametrized with error sensitivities, retention factors, and set break decay factors for each state (two states in Model 1, three states in Model 2). Furthermore, because error sensitivity increased with long exposures to consistent errors [17], we allowed error sensitivity to increase over time in both Models 1 and 2. Overall, Models 1 and 2 had 9 and 13 parameters, respectively. Given the high dimensionality of Model 2, we used BIC to reduce its complexity, arriving at 9 parameters (i.e., the rate at which error sensitivity increased was equal across all states, and no parameters were provided to allow for set-break decay in aim or acceleration states). As a result, Models 1 and 2 had the same number of parameters: 9. Ultimately, our ability to recover model parameters was high, according to a parameter recovery analysis (S4 Fig).

In Model 1 (Fig 4A), there was a single controller that learned from error via a fast state and a slow state. During a saccade, a fraction of the sum of these two states was expressed during

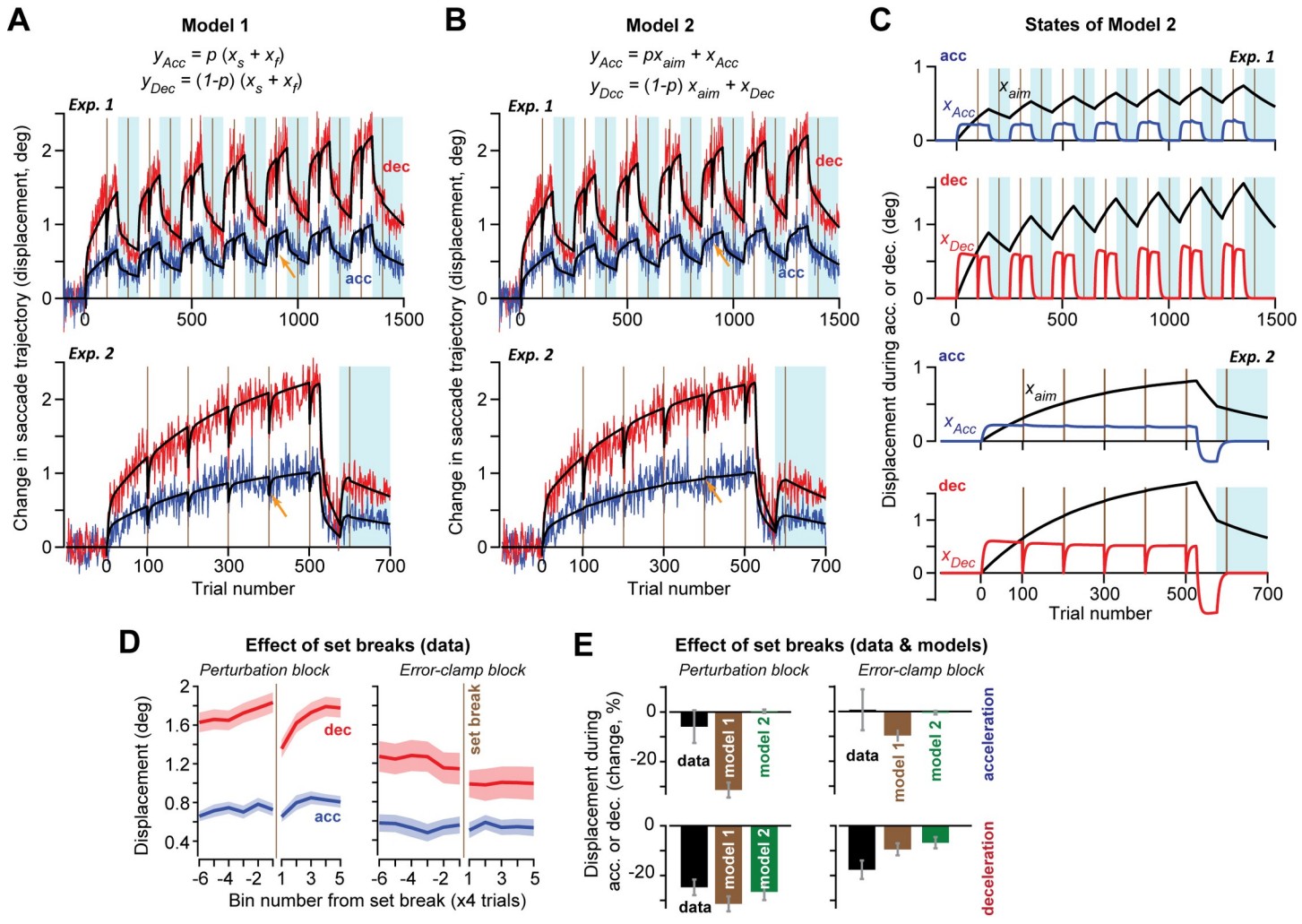

**Fig 4. Two models of adaptation. A**. Model 1 represented a single controller that learned from error via a fast state $x_f$ and a slow state $x_s$. During a saccade, a fraction of the sum of the two states was expressed during acceleration, and the remainder was expressed during deceleration. Arrow indicates the effect of set break on the model's acceleration period commands. **B**. Model 2 relied on two controllers, one that aimed the saccade with state $x_{aim}$, and another that independently affected the commands during the acceleration and deceleration periods. **C**. States of Model 2 during Exps. 1 and 2. Model 2 could account for the observation that set breaks produced a loss in the deceleration period but not the acceleration period commands. In addition, Model 2 accounted for the observation that decay following a set break differed in perturbation vs. error clamp trials: note the difference in $x_{Dec}$ at set breaks during perturbation and error clamp blocks of Exp. 1). **D**. Measured set break decay data. Adaptation is shown aligned to set breaks during perturbation periods in Exps. 1 and 2 (left) and error clamp periods in Exp. 1 (right). **E**. Comparison of the set break effects in the models and the measured data. The models differed in their ability to account for acceleration period set break decay. Error bars for data are SEM. Participants from Exps. 1 and 2 were combined when analyzing the perturbation block set breaks. For error clamp block set breaks, only participants from Experiment 1 were used.

acceleration, and the remainder was expressed during deceleration. Thus, this model suggested that faster adaptation during the deceleration period was simply because the deceleration period was longer in duration than the acceleration period. We fit the model to the data in both Experiments 1 and 2. Model 1 was able to reproduce many features of the data. For example, the deceleration period commands adapted more rapidly than the acceleration period commands. This was because in the model, only 31% of learning (sum of fast and slow states) was expressed during acceleration ($p = 0.31$, Table 1), a value that is similar to the proportion of movement duration due to acceleration (0.28). As expected, the fast state had greater sensitivity to error (as compared to the slow state), but also suffered from greater trial-to-trial

**Table 1. Parameter values and confidence intervals for Models 1 and 2.**

| Model 1 | | |
|---|---|---|
| **Parameter name** | **Mean value** | **95% CI** |
| $a_{slow}$ | 0.9963 | [0.9947,0.9996] |
| $a_{fast}$ | 0.8071 | [0.7366,0.9387] |
| $b_{slow,0}$ | 0.002852 | [0.001999,0.003714] |
| $b_{fast,0}$ | 0.03267 | [0.01228,0.04905] |
| $d_{slow}$ | 0.9481 | [0.8721,1] |
| $d_{fast}$ | 0.0599 | [$6.9 \times 10^{-6}$,0.573] |
| $p$ | 0.3127 | [0.279,0.3469] |
| $\beta_{slow}$ | $7.844 \times 10^{-6}$ | [$9.154 \times 10^{-15}$,$1.4 \times 10^{-5}$] |
| $\beta_{fast}$ | $9.397 \times 10^{-5}$ | [$3.462 \times 10^{-5}$,0.000186] |
| **Model 2** | | |
| $a_{aim}$ | 0.9817 | [0.828489,0.998409] |
| $a_{acc}$ | 0.8680 | [0.724270,0.995977] |
| $a_{dec}$ | 0.8433 | [0.748168,0.999965] |
| $b_{aim,0}$ | 0.0051 | [0.001831,0.029825] |
| $b_{acc,0}$ | 0.0063 | [0.001065,0.013130] |
| $b_{dec,0}$ | 0.0199 | [0.001520,0.031114] |
| $d_{dec}$ | 0.1680 | [0,0.861970] |
| $\beta$ | 0.0027 | [0.001004,0.004862] |
| $p$ | 0.3088 | [0.190674,0.363140] |

forgetting. Thus, Model 1 could account for the fact that both acceleration and deceleration period commands exhibited spontaneous recovery.

However, set breaks revealed a critical limitation of Model 1. Because acceleration and deceleration were derived from the same adaptive states, when adaptation decayed during deceleration, so too did adaptation during acceleration. However, in the measured data the set breaks during perturbation periods led to decay in only the deceleration period commands (Fig 4E, Exps. 1&2 deceleration commands -25 ± 3%, t(39) = -7.85, p = $1.50 \times 10^{-9}$; acceleration commands -6 ± 7%, t(39) = -0.90, p = 0.37). Similar patterns were observed in response to error clamp period set breaks (Fig 4E, data, Exp. 1; deceleration: -18 ± 4%, t(19) = -4.73, p = 0.0001; acceleration: 1 ± 8%, t(19) = 0.11, p = 0.92). In summary, while Model 1 generally fit the data well, it could not account for the differential patterns of decay that were present following set breaks.

As expected, Model 2 was able to account for spontaneous recovery, but critically, it was also able to reproduce behavior during set breaks. When set breaks occurred, acceleration period commands of Model 2 showed little decay, whereas deceleration period commands showed substantial decay (Fig 4E). The model also agreed with the data in that set breaks produced greater decay during perturbation periods than during error clamp periods (within-subject difference of 7 ± 3%, t(20) = 2.38, p = 0.027).

To understand these patterns, we examined the adaptive states predicted by Model 2 (Fig 4C). When the set breaks occurred during the perturbation block, the deceleration state was non-zero and therefore exhibited decay. However, when the set breaks occurred during the error clamp block, the same state had substantially decayed prior to the set break, thus limiting additional decay that could occur due to the set break. As a result, set breaks produced different levels of decay during perturbation and error clamp blocks.

Statistical model comparison (log-likelihood) suggested that 47 participants (73.4%) were better described by Model 2, whereas 17 participants (26.6%) were better described by Model 1 (S5A Fig). These patterns were recapitulated by a model recovery analysis (S5B Fig); when data were simulated with Model 1, log-likelihoods recovered Model 1 with about 55% probability, exceeding the rate observed in the actual data (26.6%) by about 30%. However, when data were simulated with Model 2, log-likelihoods recovered Model 2 with about 80% probability, in close agreement with the rate observed in the actual data (73.4%).

In summary, we compared two models that could potentially account for patterns of learning and spontaneous recovery. In Model 1 we assumed that there was a single adaptive controller with fast and slow states, but only a portion of its learning was expressed during acceleration, with the remainder during deceleration. In Model 2 we imagined two controllers that each had a single timescale of learning, with one controller learning to aim the saccade toward a new location, and another learning to independently modulate commands during the acceleration and deceleration periods. While both models could account for the data quite well, one critical feature of the data was more consistent with Model 2: the fact that set breaks produced forgetting in the deceleration commands, but not acceleration commands.

Note however that these findings should be interpreted cautiously, as they rely on a null result: the inability to detect set break decay in acceleration period commands. With that said, this result was robust across several analysis methods: (1) in normalized set break decay measures (Fig 1), (2) in non-normalized set break decay measures (S2B Fig), and when controlling for acceleration and deceleration magnitudes (S2C Fig). Thus, overall the data suggested that acceleration period commands exhibited a resistance to decay that was only captured by Model 2.

## Single trial learning dissociates adaptation in the acceleration and deceleration periods of a saccade

A critical element of our analysis was to divide a saccade's trajectory based on its peak velocity, producing measures of adaptation during periods of acceleration and deceleration. However, is this division warranted? That is, is there anything special about the adaptation capabilities of the commands that arrive specifically during deceleration? To answer this question, we performed a new experiment in which we aimed to emphasize adaptation in the fast states of learning, testing whether the result would dissociate the commands between acceleration and deceleration.

In Exp. 3, participants (n = 37) made primary saccades in the vertical direction (80% of the trials; Fig 5A): during the primary saccade, the target was displaced horizontally to the right or left at random and with equal probability (13% of trials in each case, Fig 5A). To measure learning from error, we considered pairs of trials, $n$ and $n+1$, in which the primary saccade was to the vertical target, but the target was jumped only in trial $n$. Learning from error was measured as the trial-to-trial change in the horizontal and vertical eye velocity ($\Delta \dot{H}$ and $\Delta \dot{V}$) of the primary saccade, that is: $\Delta \dot{H}(t) = \dot{H}^{(n+1)} - \dot{H}^{(n)}$.

As expected, there was no error-dependent change in vertical velocity (left panel of Fig 5C): in the deceleration period, there was a slight increase in vertical velocity, but the effect did not differ between $H^+$ and $H^-$ errors (peaks of 5.94 ± 0.77 vs. 5.27 ± 0.57 deg/sec, respectively, t(36) = -0.82, p = 0.42). In sharp contrast, the change in horizontal velocity following experience of a horizontal endpoint error was bimodal and differed between acceleration and deceleration. For example, following a positive error $H^+$, $\Delta \dot{H}$ was negative during the acceleration phase of the primary saccade, thus pulling the eyes in the wrong direction with respect to the error (green trace, right panel of Fig 5C). However, during the deceleration period, $\Delta \dot{H}$ became positive,

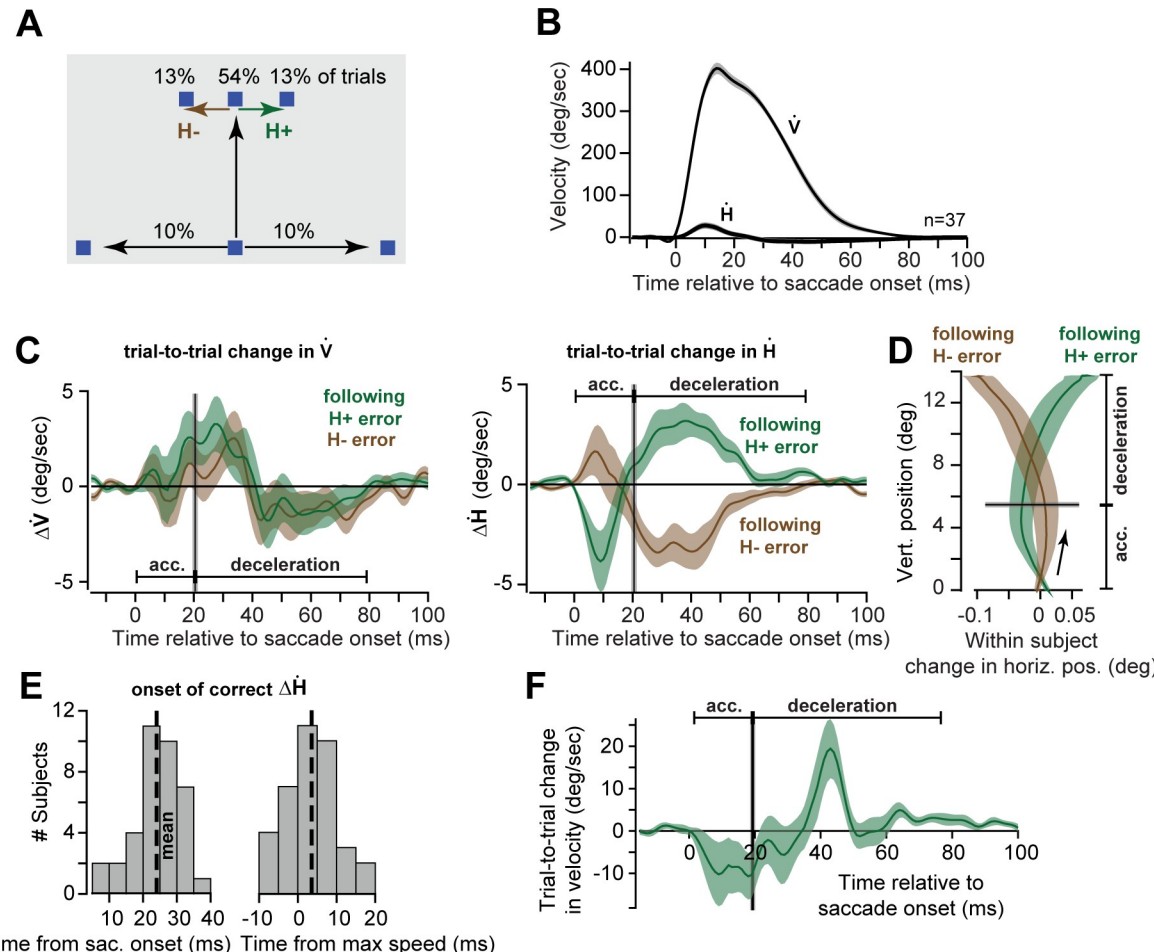

**Fig 5. Experiment 3.** Following experience of a single error, correction of the subsequent saccade was primarily during deceleration. **A**. Subjects made vertical saccades and occasionally experienced a horizontal error. **B**. Average horizontal and vertical velocity of the saccade for the target along the vertical axis. **C**. Trial-to-trial change in vertical and horizontal velocity following experience of a positive or negative horizontal endpoint error. Vertical line indicates time of peak speed, thus separating the acceleration and deceleration periods. **D**. Trial-to-trial change in saccade trajectory following experience of a positive or negative horizontal endpoint error. The endpoint correction was primarily due to commands in the deceleration period. Note the large difference in the scales of the x and y axes. **E**. Onset time of correction in the horizontal velocity following experience of a horizontal endpoint error. Mean onset time is about 3 ms following start of the deceleration period. **F**. Data from Experiments 1 and 2, measuring the response following the experience of error on the very first trial (velocity on trial 2 minus trial 1, in the direction of the perturbation). Shaded error regions are between subject SEM.

pulling the eyes in the direction that compensated for the error. A similar bimodal pattern was present following experience of $H^-$ error (brown curve, right panel of Fig 5C).

The effect of this one-trial learning on eye trajectory is displayed in Fig 5D. Experience of an error to the right ($H^+$, green trace, Fig 5D) was followed by a primary saccade that during acceleration changed the eye trajectory in the wrong direction; however, as the saccade continued into the deceleration phase, the eye was steered correctly, thus partially compensating for the error experienced in the previous trial. As a result, following an error the next saccade improved in the sense that its endpoint exhibited a smaller error (or would have, had the perturbation repeated). Notably, this reduced endpoint error was due to commands that arrived solely during the deceleration period.

To quantify the exact time that the commands correctly changed the eye's trajectory, we analyzed data of each participant separately and looked for the onset time in which $\Delta\dot{H}$

corrected the eye trajectory in the direction of the previous error. The results revealed that error compensation did not begin at saccade initiation (Fig 5E, left subplot, mean 24.14 ± 1.12ms, t(36) = 21.54, p = 7.38X10$^{-22}$). Rather, the compensation started 3.67 ± 1.05ms following peak velocity, i.e., slightly after the onset of deceleration (right subplot, Fig 5E).

To check the robustness of this result, we re-analyzed the data in Experiments 1 and 2 (n = 64 subjects). We measured the change in eye velocity (in the direction of the perturbation) from the first perturbation trial to the next trial only. We found that following experience of an error in trial 1, the correction for that error in trial 2 occurred only during the deceleration period (Fig 5F). It was noteworthy that this adaptive response had a larger magnitude than we had observed in Exp. 3. This suggests that when errors are random and reverse direction (as in Exp. 3), there may be a suppression of the adaptive response to error (as compared to errors that are consistently in the same direction) (Herzfeld et al., 2014).

In summary, following experience of an error, the subsequent primary saccade included small, corrective commands that steered the eyes to reduce the error. However, the expression of these corrective commands did not begin until the onset of the deceleration period (thus violating the coupling required in Model 1). One should always be cautious about interpreting null results such as the lack of adaptation during the acceleration period; however, this result was consistent across all three experiments, bolstering its credibility. Overall, we found that the adaptive response to the experience of a single error was present only during deceleration period of the subsequent saccade.

## Discussion

Spontaneous recovery of motor memory is an interesting phenomenon that was correctly predicted by the two-state model of adaptation [8]. Despite the simplicity of the model, however, it has been difficult to identify behavioral correlates of the putative fast and slow states. Neural correlates of the fast and slow processes have been found in the synaptic plasticity mechanisms of single neurons or groups of neurons in the cerebellum [3,18,18–23]. Correlates of the fast and slow processes have also been noted in distinct regions of the cerebral cortex and the cerebellum [24–37]. Finally, different memory systems (explicit and implicit) have been proposed to contribute to the two putative processes [38,39]. Given these multitude of neural correlates, might it be possible to find behavioral signatures as well? Here, we aimed to identify such correlates in the commands that guide saccadic eye movements.

In Experiments 1 and 2 we observed that following experience of an error, motor commands that guided the subsequent saccades were altered. However, while the acceleration period commands learned relatively little from error and exhibited little forgetting during the set breaks, the deceleration period commands not only adapted with high sensitivity to error, they also suffered significant forgetting during set breaks. These patterns suggested that perhaps the acceleration period commands reflected a slow state of learning, whereas the deceleration period commands reflected a fast state. However, the predictions of this simple model were inconsistent with the data during spontaneous recovery: when adaptation was followed by extinction, the motor commands during the acceleration and deceleration periods both showed spontaneous recovery in the subsequent error clamp block. Thus, the simple two-state model could not account for properties of saccade trajectory during adaptation and spontaneous recovery.

To better understand the experimental results, we considered two models. In Model 1, we imagined that learning was due to a fast and a slow process, but that during a movement some fraction of each process would be expressed during the acceleration period, and the remainder during deceleration. This is akin to a scenario in which learning is limited to a forward model,

which corrects the planned movement as it is executed [16]. In Model 2, we imagined two controllers that each had a single timescale of learning, with one controller learning to aim the saccade toward a new location, and another learning to independently modulate commands during the acceleration and deceleration periods. While both models produced good fits to the data, Model 1 could not account for the result that set breaks produced forgetting in only the deceleration period commands. Thus, the modelling work suggested that learning from error engaged two distinct adaptive processes, providing some independence between the adaption that was expressed during acceleration and deceleration.

To more cleanly test the coupling between acceleration and deceleration periods, we performed Experiment 3, in which the participants experienced a random horizontal error following the conclusion of a vertical saccade. This error induced a change in the commands that guided the subsequent vertical saccade, partially correcting for the error. However, the error-dependent correction was not uniformly expressed throughout the saccade. Rather, the correction took place only via commands that arrived during the deceleration period, providing key support for the idea that the deceleration period commands could adapt independently of acceleration.

There are limitations in our modeling. First, the delineation between Models 1 and 2 is primarily driven by differences in time-based decay, which itself is based on an inability to detect set break decay in acceleration period commands. Second, we encountered a puzzle in Exp. 3 that we could not explain with our models: following experience of an error, the subsequent saccade had deceleration period commands that partially corrected for the previous error. However, there were also changes in the acceleration period commands that moved the eyes in the wrong direction with respect to that error. We currently have no explanation as to why a random sequence of errors should produce such changes in the acceleration period commands of saccades.

Our division of saccade trajectory into acceleration and deceleration was based on our previous observations regarding curvature of saccades during cross-axis adaptation [16]. In that work we found that when a saccade path was divided into four equal segments, the slopes of the first two segments were roughly equal, suggesting a straight line. However, the slopes doubled by the fourth segment, exhibiting a non-linear increase that started at around saccade midpoint, shortly after peak velocity. Results of Exp. 3 in our current report illustrated that while the adaptive response to a single error was present after peak velocity of the saccade, it was missing before this time point.

Taken together, the data suggest that the adaptive response to error is different during the acceleration and deceleration periods: 1) the deceleration period commands learn more from error, and also show a greater forgetting, and 2) adaptation following exposure to a single error specifically engages the deceleration period.

## Potential links to neural mechanisms of adaptation

Control of saccade trajectory depends on an internal feedback process that monitors the ongoing movement [40,41], resulting in "steering" of the saccade. For example, in humans it is possible to disrupt a saccade mid-trajectory by using a single pulse of a transcranial magnetic stimulation. Remarkably, that saccade is corrected within the same movement [42]. This internal feedback process likely depends on the integrity of the cerebellum [1,2,16], suggesting that as a saccade is generated by the superior colliculus, the cerebellum monitors and corrects the ongoing motor commands.

Here, our data suggested a degree of independence between the acceleration and deceleration period commands. One possibility is that changes in the acceleration period commands

are a form of re-aiming the saccade (Model 2). Re-aiming is present in reach adaptation studies [43–46], raising the possibility that it may also be present in saccade adaptation. If so, we would expect that saccade adaptation would involve changes in how the superior colliculus responds to the visual target. The possibility of target remapping was studied by Quessy et al. [47] and Frens and Van Opstal [48]. They recorded from the superior colliculus and found that during saccade adaptation, locus of activity on the superior colliculus appeared unchanged, thus making remapping unlikely (but see Takeichi et al. [49]). However, Gaymard et al. [50] reported reduced saccade adaptation in patients with lesions in the cerebellar thalamus. They suggested that the thalamus relayed adaptation-related information from the cerebellum to the cortical oculomotor areas, which can in principle remap the target. Thus, if there is remapping of the target during saccade adaptation, the changes in acceleration period commands might be due to this pathway.

To test whether saccade adaptation involves target re-mapping, future studies can borrow an idea from reach adaptation and manipulate reaction times [43,51]. The idea is that target remapping requires computation time for the brain, and thus by extending reaction times, the brain has a greater potential to re-aim the movement. According to this framework, saccades that have short reaction times, for example express saccades, should exhibit little to no re-aiming, whereas saccades that have long reaction times, for example memory guided saccades, should exhibit significant re-aiming.

Saccades that we studied here are termed "reactive saccades", movements that are triggered in response to sudden appearance of a visual stimulus. Adaptation of reactive saccades relies on the cerebellum [1,52,53]. However, adaptation of voluntary saccades, in which movements scan a scene, relies on both the cerebellum [54] and other structures such as the parietal cortex [55]. It is possible that voluntary saccades in the context of cross-axis adaptation will exhibit a greater amount of re-aiming, and thus produce a larger change in the acceleration period commands.

It seems likely that changes in the deceleration period commands are at least partly due to an adaptive response from the cerebellum. Cerebellar Purkinje cells (P-cells) may be organized into groups based on their preference for direction of visual error [21,56]. This preference for error is expressed via tuning of complex spike probability with respect to the direction of the visual error [21,56,57]. When a saccade concludes and there is a visual error, the result is an increase in the probability of complex spikes for some P-cells, and a decrease in complex spike probability for other P-cells. Both the increase and the decrease in complex spikes promote plasticity in the P-cells: depression of parallel fiber synapses follows presence of the complex spike, and potentiation follows its absence [3,23,58]. However, the plasticity is greater following presence of a complex spike. Thus, experience of a single error may be followed by differing amounts of learning among various P-cells. This hints that the multiple timescales of adaptation, as expressed in behavior, may be at least partly due to the diverse responses of various P-cell populations to a single error [21].

The idea that there may be separate adaptive controllers for acceleration and deceleration may be relevant to reaching: one controller learns to re-aim the reach, whereas another corrects the movement as it takes place. However, single trial learning in reaching exhibits an error-dependent response that is different than what we found during saccades. In saccades, learning from a single error resulted in corrective motor commands that were expressed primarily during the deceleration period. In reaching, experience of an error produced corrective motor commands that were expressed throughout the subsequent reach [59]. That is, unlike saccades, the single trial response to error for reaching adjusts the motor commands during the entire movement.

Overall, our results demonstrate that the deceleration phase of a saccade exhibits adaptation properties that are distinct from the acceleration phase. This raises the possibility that different

neural adaptive control mechanisms contribute to the acceleration and deceleration phases of a single saccade.

## Methods

### Ethics statement

All subjects gave written informed consent in accordance with policies approved by the Johns Hopkins School of Medicine Institutional Review Board (Protocol Number: NA_00087187).

Our goal was to ask whether there were behavioral correlates of the fast and slow adaptive processes within a single movement. In order to do so, we focused on cross-axis saccade adaptation [2,16]: in all experiments, the primary target was placed at 15˚ from the fixation point at the center of the screen, and upon saccade initiation (on perturbed trials), the target was erased and replaced with a new target at 5˚ perpendicular to the direction of the original target (Fig 1A). Thus, the primary movement direction and direction of error were perpendicular. This approach has its roots in force field reaching experiments [60], and has the advantage that it disambiguates the primary movement commands from the commands that change in response to error.

### General protocol

Participants sat in a chair with their head restrained by a bite-bar made from dental putty (Coltene-Whaledent Inc. Cuyahoga Falls, Ohio, USA). They viewed a computer monitor (AGON 27" 2560X1440, 144Hz) which displayed small square targets (0.25˚ with 0.1˚ dark border). They made saccades in response to presentation of the target, and eye movements were recorded from the right eye at 1000 Hz using Eyelink 1000 (S-R Research Ottawa, Ontario, CA).

The basic trial structure is shown in Fig 1A. Each trial began with a fixation period of variable duration (randomly sampled from a uniform distribution 250-750ms), during which subjects had to maintain fixation in the center of the screen within a tolerance window of ±2.5˚. If they blinked or looked away early, the timer was reset and the trial repeated. After the fixation period, the fixation target disappeared and was replaced by the saccade target. On perturbed trials, immediately after detection of saccade onset the target was jumped 5˚ perpendicular to the primary saccade direction. Subjects were required to fixate the final target location for 250 ms to complete the trial.

The experiments started with a baseline period without target jumps. During the baseline block, we measured saccade kinematics to targets that were randomly presented on the vertical or horizontal axes, as well as targets along various oblique directions. Vertical saccades were made to a target at (0,+15˚). Horizontal saccades were made to targets at (±15,0˚). Oblique saccades were made to targets at (±1,15˚), (±2,15˚), (±3,15˚), (±4,15˚), or (±5,15˚). The oblique saccades served as a control so that we could compare saccade kinematics following adaptation with baseline saccades that had the same endpoint (see Fig 2).

In error clamp trials, our objective was to place the target at the location where the saccade would end. To do so, we used linear extrapolation from the state of the eye near saccade end to predict where the eyes would land at the conclusion of the saccade. We measured eye position, velocity, and acceleration when saccade speed fell below 150 deg/sec and used extrapolation to predict gaze location at the time when velocity would cross 0˚/sec in the horizontal and vertical directions independently. We then placed the target at that location. To quantify the effectiveness of this approach, we compared the residual error, defined as eye position at saccade end with respect to target, in error clamp trials to error in baseline trials in which the target was not moved. We found that during error clamp trials, errors at saccade endpoint were no different than those observed during baseline saccades (S6 Fig).

## Experiment 1

In this experiment we collected data from 26 subjects, but 2 were removed due to poor calibration that resulted in missing data for $\geq$15% of trials (final n = 24, 22.42 ± 4.0 years old, 11 females). The perturbation was always +5˚ in the horizontal axis on upward vertical trials (Fig 1A). Data from randomly inserted horizontal trials (20% of trials) were not analyzed; these trials were included to introduce uncertainty about the primary saccade location and encourage subjects to maintain fixation for the entire fixation period.

Following a baseline period of 200 trials (80 vertical, 80 oblique, and 40 horizontal), adaptation periods were alternated with error clamp periods, as shown in Fig 1C. Block 1 was an adaptation period of 150 trials (120 vertical trials, 30 horizontal trials), and the subsequent blocks included 100 trials each (80 vertical trials, 20 horizontal trials) until the final error clamp period (150 total trials, 120 vertical trials). In addition, the experiment included set breaks of 30 sec in duration.

The transitions between adaptation and error clamp periods happened in the middle of each block rather than at set breaks to ensure that decay during set breaks did not mask or interfere with early trial-to-trial decay in the error clamp periods. Total experiment length was 1700 trials (1280 vertical trials, 340 horizontal trials, and 80 oblique trials), with 7 adaptation periods and 7 error clamp periods.

## Experiment 2

We collected data from n = 40 subjects, 26.95 ± 4.86 years old, 10 females. Following a baseline period as in Experiment 1, subjects experienced a 5˚ perturbation (Fig 3C). As before, the perturbation direction was always perpendicular to the primary target direction. However, the subjects were divided into two equal groups in which the primary target was either upward (vertical) or rightward (horizontal). These two groups were further subdivided equally based on the initial direction of the perturbation (rightward vs. leftward for the vertical group, upward vs. downward for the horizontal group) for a total of four groups with 10 subjects each. The initial perturbation direction was maintained for 525 total trials (420 perturbed trials, 125 perpendicular trials, similar to the horizontal trials in Exp. 1), after which the perturbation switched direction, as shown in Fig 3A, lower subplot: that is, the initial perturbation trials were followed by a consistent 5˚ perturbation in the opposite direction for 50 total trials (40 perturbed trials). The experiment concluded with a block of error clamp trials (125 trials) to assess spontaneous recovery.

## Experiment 3

Here we aimed to examine the learning response to a single error. The setup is shown in Fig 5A. We collected data from 38 subjects, but one was excluded due to poor calibration that made over 25% of trials unusable (final n = 37, 26.08 ± 8.9 years old, 22 females). The experiment consisted of 1350 total trials. The baseline period (180 trials) consisted of 64 trials in which the target was presented along the vertical axis (0,+15˚), 36 trials in which the target was presented along the horizontal axis (±15,0˚), and 80 trials in which the target was in an oblique direction (as described above). In the perturbation block, rightward and leftward perturbations (±5˚) were randomly interleaved with control (no jump) trials to vertical targets (0,+15˚) so that every perturbation trial was preceded and followed by at least one control trial to ensure that the measured trial-to-trial changes were truly due to a single error. There was a total of 156 rightward (+5˚) and 156 leftward (-5˚) target jumps in the perturbation period. We noted a puzzling effect of small changes in the initial horizontal position of the saccade on the acceleration phase of the movement (Figs 5D and S7); therefore, we excluded movements for which

the change in horizontal position was greater than 0.5 in either direction. On average, 35 trials (12%) per subject were thus excluded. We divided each saccade into an acceleration phase and a deceleration phase based on peak speed (magnitude of the velocity vector), which was largely dominated by the vertical component, and examined the trial-to-trial change during each movement phase in response to a single error.

## Data analysis

We analyzed our data using in-house code written in MATLAB (The Mathworks, Natick, MA USA), except the repeated measures ANOVAs, which were performed in Rstudio for R using the aov function. Position and velocity data from the eye were filtered off-line using a $3^{rd}$-order low-pass Butterworth filter with a cutoff frequency of 200 Hz prior to analysis.

For Experiments 1 and 2, we averaged data in bins of 4 trials and then performed a within-subject repeated measure ANOVA with trial bin and movement phase (acceleration and deceleration) as main effects, including the interaction term, and displacement in the adapted axis as the outcome measure. Learning rates were computed as the mean increase in adaptation during acceleration and deceleration divided by the number of trials (Fig 1E). Loss due to set break was computed by aligning displacement data for each subject to set breaks (separately for perturbation and error clamp periods) and then averaging across set breaks to produce a single subject response (Fig 1F). We then computed the within subject percent change in displacement due to the set breaks and performed statistics on this measure. Subjects who failed to reach at least 0.25˚ of adaptation in both phases of the movement prior to the set break were excluded from the percent change analysis (4 subjects thus excluded for Exp. 1 set breaks). The reported changes in horizontal and vertical velocity during the first set break are the average difference between the last bin prior to the set break and the first bin following the set break during each phase of the movement.

To measure decay patterns during error clamp periods (Fig 1G), we averaged the data of each subject within bins of 4 trials each, aligned it to the transition into error clamp from the perturbation period, and computed the average response. We then quantified the within subject percent change in saccade displacement in the error clamp period with respect to the last bin of the preceding perturbation period. We performed a within-subject repeated measure ANOVA for the percent retention with main effects of movement phase and trial bin as well as their interaction (Fig 1G).

As a control for Experiment 1, we compared the relative contributions of the acceleration and deceleration phases to total horizontal displacement for baseline (control) oblique saccades and adapted saccades with similar amounts of total horizontal displacement (Fig 2). We first performed a 2-way within subject ANOVA with main effects of movement phase and condition (control vs. adapted) including the interaction term. We further calculated the percent contribution of acceleration and deceleration collapsed across the 3 levels of horizontal displacement (1˚, 2˚, and 3˚) and performed paired two-tailed t-tests.

We assessed potential differences in groups for Experiment 2 (S3 Fig) using a mixed-model ANOVA with error direction (horizontal vs vertical) as the single between subject factor and trial bin number (each bin is 4 trials) and movement phase (acceleration vs. deceleration) as the 2 within subject factors.

We quantified spontaneous recovery by using three periods of 8 trials each (labeled in Fig 3B and 3C), where $t_1$ is the end of the initial adaptation period, $t_2$ is the end of the extinction period, and $t_3$ is the peak of the spontaneous recovery curve (immediately after the final set break). We defined spontaneous recovery as the difference between adaptation at $t_3$ and $t_2$ divided by the adaptation at $t_1$ (Fig 3E), that is: $[x(t_3)-x(t_2)]/x(t_1)$. Subjects whose adaptation at

$t_1$ in either movement phase was lower than 0.25˚ were excluded from the spontaneous recovery analysis (5 subjects were thus excluded, final n = 35 for spontaneous recovery).

Trial to trial analysis for Experiment 3 relied on the endpoint error that the subjects experienced on any given trial. We calculated error as the final position of the target minus the position of the eye at saccade end and defined errors less than -2.5˚ as negative errors (H-), and errors of greater than +2.5˚ as positive errors (H+). We then analyzed the change in horizontal position, displacement, and velocity, from the trial in which the error was experienced to the next trial, as well as the change in vertical velocity (Fig 5C and 5D). To compute the time during the saccade in which the motor commands corrected for the previous error, we flipped the sign of the response to H- errors and combined them with the responses to H+ errors to obtain a single response to error profile for each subject. We then followed the procedures described by [61] to find the timing of the inflection point in the trajectory, resulting in the data shown in Fig 5E.

## State-space models

After experience of an error, humans and other animals change their behavior on the subsequent trial. In the absence of error, the adapted behavior decays over time. We used a state-space model to capture this process of error-based learning in our experiments [8,62,63]. The state-space model describes behavior as the summation of underlying hidden states. We represented the internal states via the vector *x*. We investigated the possibility that different parts of the same movement are supported by distinct states, focusing specifically on the acceleration and deceleration components of a single saccade.

One hallmark of adaptation is the presence of spontaneous recovery, a phenomenon in which past learning can be re-expressed in the absence of error. This process requires the existence of multiple timescales of memory, at least one fast and one slow state in the adaptive process [8,15]. In Experiment 2, we found evidence for spontaneous recovery in cross-axis adaptation of saccades. To describe this spontaneous recovery, we first used a standard two-state model. Our initial hypothesis was that the deceleration phase would resemble the model's fast state, and the acceleration phase would resemble the model's slow state. This turned out not to be true, as we found recovery in both the acceleration and deceleration phases of movement, suggesting that both periods have contributions from slow and fast states. Thus, we instead developed two alternate models that could account for these data: Models 1 and 2.

Model 1 was similar to a standard two state model in which there was a single controller that adapted with two states and the sum of these two states affecting behavior. However, a fraction of adaptation (sum of fast and slow states) was expressed early in the movement (acceleration period), and the remaining fraction was expressed later in the movement (deceleration period). Thus, in this model learning during acceleration and deceleration was derived from a single source.

In Model 2, there were two controllers that contributed to adaptation. One controller produced commands which re-aimed the saccade and contributed to both acceleration and deceleration period commands. A second controller produced commands which independently altered the acceleration and deceleration periods. Here, the idea was that there was a system that learned to aim the saccade, and another system that corrected that movement as it took place. Model 2 was inspired by physiological data regarding control of saccades by the superior colliculus and the cerebellum, which suggested that the cerebellum may play a particularly important role for steering the eyes during the declaration period of a saccade that was initiated by the colliculus. Therefore, in this model, learning during acceleration and deceleration was derived from different sources.

Below, we first describe the elements of our state-space framework common to each model. Then we describe our initial hypothesis, fitting a standard two-state model to the data. Finally, we describe Model 1 and Model 2 each in more detail.

## Overview of the state-space model

In a standard state-space model of adaptation, the internal states of the learner change from trial $n$ to trial $n+1$. This change is due to learning from error, $e^{(n)}$, and trial-by-trial decay:

$$\boldsymbol{x}^{(n+1)} = A\boldsymbol{x}^{(n)} + be^{(n)} \tag{1}$$

In Eq (1), $x$ is a vector that contains the state of adaptation. Forgetting is controlled by the retention matrix $A$, a diagonal matrix which enforces that each state independently decays exponentially over trials. The rate of learning is controlled by the error sensitivity vector $b$. The error sensitivity vector consists of the individual error sensitivities for each internal state. In some cases (see Models 1 and 2 description below) error sensitivity can increase over time, leading to savings (Herzfeld et al. 2014).

In biological terms, the error is a vector that reflects position of the target with respect to the fovea in the post-saccadic period [21]. This is inferred by the fact that low-intensity stimulation of the superior colliculus at around 80 ms following saccade completion produces complex spikes in the cerebellum [64], and is followed by behavioral changes that resemble saccade adaptation [65,66]. Here, the perturbation was perpendicular to the direction of the main target, and thus we represented error as a scalar quantity that was the distance between saccade endpoint and target position following saccade end.

While we cannot directly measure the internal states $x$, we can measure their combined effect on the adapted movement:

$$\boldsymbol{y}^{(n)} = C\boldsymbol{x}^{(n)} \tag{2}$$

In the above equation, $C$ is a matrix that describes the relationship between the measured behavior and the adaptive states. Here, we describe the behavior, $\boldsymbol{y}$, as a vector that consists of two components: horizontal displacement during acceleration, $y_A$, and horizontal displacement during deceleration, $y_D$. The sum of these two displacements yields the total horizontal displacement of the eye, $y$.

Our experiments consisted of a combination of trial conditions. On some trials, a perturbation $r$ was applied to the target position. On these trials, the error experienced by the participant was equal to the difference between the perturbation, and the total displacement of the eye. On error clamp trials, movement errors were eliminated. Therefore, the error in Eq (1) has two possible forms:

$$e^{(n)} = \begin{cases} r^{(n)} - y^{(n)}, & \text{not an error clamp trial} \\ 0, & \text{error clamp trial} \end{cases} \tag{3}$$

Lastly, to model the effect of set breaks (Albert & Shadmehr, 2018), we allowed for decay in the internal states via the matrix $D$:

$$x^{(n+1)} = D^{(n)}\left(Ax^{(n)} + b^{(n)}e^{(n)}\right) \tag{4}$$

We allowed decay to differ for each internal state. On normal trials not followed by a set break, the $D$ matrix was set equal to the identity matrix. Therefore, our final state-space model

is given by:

$$x^{(n+1)} = D^{(n)}(Ax^{(n)} + be^{(n)})$$
$$y^{(n)} = Cx^{(n)}$$

(5)

The form of $A$, $b$, $C$, $D$, and $x$ differ for each model. In the following sections, we describe each of these models in more detail.

## The two-state model

We initially tested the possibility that the fast-adaptive state was expressed during deceleration and the slow-adaptive state was expressed during acceleration. Thus, we fit a standard two-state model to our data using the methods initially outlined by [8]. In this model, the total displacement of the eye is described, rather than the individual acceleration and deceleration components. Thus, this model is equivalent to Eqs (3)–(5) with the exception that $y$ is a scalar (a total displacement) rather than a two-dimensional vector (an acceleration and deceleration component). In this case, the internal state vector $x$ consisted of a slow and fast state: $[x_s \, x_f]^{\mathrm{T}}$. The state vector learned and decayed according to the error sensitivity vector $b$ and the retention matrix $A$. The vector $b$ was specified by $[b_s \, b_f]^{\mathrm{T}}$. The retention matrix $A$ was specified by a 2 x 2 diagonal matrix whose main diagonal consisted of the retention factors $a_s$ and $a_f$. Finally, the set break decay matrix $D$ was also a 2 x 2 diagonal matrix whose main diagonal consisted of parameters $d_s$ and $d_f$ which represent the fraction that the slow and fast state decay with the passage of time during a set break (Eq (4)).

We fit this model to the mean behavior measured in Experiment 2 (shown in Fig 3B). In this fit, we enforced the standard two-state dynamics, namely, that the fast state is more sensitivity to error ($b_f \geq b_s$) and the slow state retains its memory more robustly ($a_s \geq a_f$). To fit the model to behavior, we minimized the squared error between the model and the average data across all participants in Experiment 2. The fast and slow states predicted by the model are shown in Fig 3C. Note that these states did not resemble the deceleration and acceleration commands shown in Fig 3B.

## Model 1

In our first model, we imagined that the adaptation expressed during acceleration and deceleration was derived from a single source. As time elapsed during a movement, a fraction of the adapted motor commands was expressed during the beginning phase of the movement (i.e., acceleration) and the rest of adaptation was expressed during the later phase of the movement (i.e., deceleration). Therefore, Model 1 supposed that $y_A$ and $y_D$ were derived from the same states and each differentially expressed some fraction of the total adaptation. In this case, Eq (2) takes the following form:

$$y_A = p(x_s + x_f)$$

$$y_D = (1 - p)(x_s + x_f)$$

(6)

Here the variable $p$ represents the fraction of total adaptation expressed during the acceleration phase. By extension, the quantity $1 - p$ represents the fraction expressed during deceleration. Adaptation from both phases is derived from the same slow and fast states, denoted $x_s$ and $x_f$, respectively.

The slow and fast states learn and decay (as in Eq 1) using the standard form of the error sensitivity vector $b$ and retention matrix $A$. The vector $b$ consists of the error sensitivity for the

slow, $b_s$, and fast, $b_f$ states of learning. The retention matrix $A$ is a 2 x 2 diagonal matrix whose main diagonal consists of the retention factor for the slow, $a_s$, and fast, $a_f$, states of learning.

Critically, because Experiments 1 and 2 involved long and sometimes repeated (Exp. 1) exposure to the perturbation, we allowed error sensitivity to increase over time [17]. Without this, the state-space model cannot exhibit a key aspect of adaptation: savings, i.e., a faster and more complete learning with each re-exposure to the perturbation. Thus, in the model, $b_s$ and $b_f$ both start at initial values of $b_{s,0}$ and $b_{f,0}$ respectively, and increase by $\beta_s$ and $\beta_f$ on each perturbation trial (i.e., error sensitivity increased only when an error was observed, not on error clamp trials). Note that because $b$ is specific to a given error [17], $b_s$ and $b_f$ were reset in Experiment 2 back to $b_{s,0}$ and $b_{f,0}$ when the perturbation switched sign.

We enforced the conventional two-state dynamics, namely, that the fast state is more sensitive to error ($b_{f,0} \geq b_{s,0}$) and the slow state retains its memory more robustly ($a_s \geq a_f$). Finally, the set break decay matrix $D$ is also a 2 x 2 diagonal matrix whose main diagonal consists of parameters $d_s$ and $d_f$ which represent the fraction that the slow and fast state decay with the passage of time during a set break (Eq (4)).

In summary, Model 1 consists of 9 parameters: {$a_s$, $a_f$, $b_{s,0}$, $b_{f,0}$, $d_s$, $d_f$, $p$, $\beta_s$, $\beta_f$}. We will describe the process by which we fit the model to behavior in text below.

## Model 2

In this model, we imagined two controllers that each had a single timescale of learning. One controller learned to aim the saccade toward a new location, and another learned to independently modulate commands during the acceleration and deceleration periods. The aiming controller re-aimed during acceleration by the proportion $p$ and deceleration by the proportion $1 - p$:

$$y_{acc}^{(n)} = p x_{aim}^{(n)} + x_{acc}^{(n)}$$

$$y_{dec}^{(n)} = (1 - p) x_{aim}^{(n)} + x_{dec}^{(n)} \tag{7}$$

The states changed following experience of error:

$$x_{aim}^{(n+1)} = a_{aim} x_{aim}^{(n)} + b_{aim} e^{(n)}$$

$$x_{acc}^{(n+1)} = a_{acc} x_{acc}^{(n)} + b_{acc} e^{(n)}$$

$$x_{dec}^{(n+1)} = a_{dec} x_{dec}^{(n)} + b_{dec} e^{(n)} \tag{8}$$

Therefore, the internal state vector $\mathbf{x}$ consisted of $[x_{aim}\, x_{acc}\, x_{dec}]^{\mathrm{T}}$. The state vector learned and decayed according to the error sensitivity vector $\mathbf{b}$ and retention matrix $A$. The vector $\mathbf{b}$ was specified by the vector $[b_{aim}\, b_{acc}\, b_{dec}]^{\mathrm{T}}$. The retention matrix $A$ was a 3 x 3 diagonal matrix whose main diagonal consisted of the retention factors $a_{aim}$, $a_{acc}$, and $a_{dec}$. Finally, each state decayed independently during a set break according the 3 x 3 diagonal matrix $D$, whose main diagonal consisted of the time-decay parameters $d_{aim}$, $d_{acc}$, $d_{dec}$. Similar to Model 1, to permit savings we allowed error sensitivity to increase over time. To do this, $\mathbf{b}$ started at initial values {$b_{aim,0}$, $b_{acc,0}$, $b_{dec,0}$} and increased on each perturbation trial. As in Model 1, $\mathbf{b}$ was reset to its initial values when the perturbation switched sign in Experiment 2. To allow error sensitivity to increase over time, we multiplied the initial error sensitivity vector element-wise by scaling factor determined by {$\beta_{aim}$, $\beta_{acc}$, $\beta_{dec}$}. Given the high dimensionality of this parameter set, we conducted a set of model comparisons to determine which, if any, of the trial-based retention

(a), time-based retention (d), and error-based learning parameters (b and β) were common across the various states.

To compare these model variants, we fit each to participant measured data in Experiments 1 and 2 by minimizing the following cost function:

$$\theta_{fit} = \underset{\theta}{argmin} \sum_{n=1}^{N} (y_A^{(n)} - \hat{y}_A^{(n)})^2 + (y_D^{(n)} - \hat{y}_D^{(n)})^2 \qquad (9)$$

Here, $\hat{y}_A$ and $\hat{y}_D$ refer to the amount of adaptation expressed during acceleration and deceleration predicted by each model (Eq (7)). Therefore, rather than fit the model to the total displacement of the eye, we fit the model to the individual acceleration and deceleration commands observed on each trial. To do this, we performed 100 iterations of *fmincon* in MATLAB R2018a (subject to the inequality constraints describing the conventional two-state dynamics), each time varying the initial parameter guess. We selected the parameter set across the 100 iterations that minimized squared error.

After fitting each model variant, we used the model's RMSE, the number of observations for each participant, and the number of free parameters to compute the BIC for each model. We did this for each participant, and then added each BIC across participants to calculate the total BIC for each model. This model comparison revealed that the most parsimonious model variant that best explained the behavior in Experiments 1 and 2 required only the following 9 parameters: $\{a_{aim}\ a_{acc}\ a_{dec}\ b_{aim,0}\ b_{acc,0}\ b_{dec,0}\ d_{dec}\ \beta\ p\}$. The $\beta$ represents that all 3 states increased their error sensitivity at the same rate.

## Behavioral and statistical comparison of Models 1 and 2

In Fig 4 we investigated whether Models 1 and 2 could account for various aspects of behavior measured in Experiments 1 and 2. We compared the models in two different ways. First, we focused on how each accounted for key behavioral findings, such as the magnitude of decay after a set break. Second, we compared each model statistically via maximum likelihood (S5A Fig).

To compare the behavioral phenomena predicted by Models 1 and 2, we identified the model parameters that best fit the measured behavior in Experiments 1 and 2 (Fig 4A) by minimizing the following cost function:

$$\theta_{fit} = \underset{\theta}{argmin} \sum_{n=1}^{N} (y_{A,E_1}^{(n)} - \hat{y}_{A,E_1}^{(n)})^2 + (y_{D,E_1}^{(n)} - \hat{y}_{D,E_1}^{(n)})^2 + (y_{A,E_2}^{(n)} - \hat{y}_{A,E_2}^{(n)})^2 + (y_{D,E_2}^{(n)} - \hat{y}_{D,E_2}^{(n)})^2 \quad (10)$$

Here, the *y* terms refer to either the mean acceleration and deceleration measured in Experiment 1 or Experiment 2 (denoted with the subscripts $E_1$ and $E_2$ respectively). The $\hat{y}$ terms denote the behavior predicted by either Model 1 or Model 2. We chose this cost function so as to maximize the robustness of our least-squares fitting technique, which is prone to errant predictions when measured behaviors are described by multiple hidden components [67]. Specifically, we fit to Experiments 1 and 2 simultaneously to ensure generalization of the underlying parameter set. We also fit to the mean behavior across participants (all participants that were adapted to horizontal errors), to reduce the impact of behavioral noise sources on the fitting process. And finally, we fit not to the total displacement, but to the individual acceleration and deceleration components to further constrain the model fit. Under these conditions, we expect our least-squares algorithm to perform similarly to other maximum likelihood techniques [67]. To further assess our ability to accurately recover model parameters, we also conducted a parameter recovery analysis (S4 Fig).

This minimization was performed using *fmincon* in MATLAB R2018a. To ensure that the global minimum of Eq (9) was identified, we performed 100 iterations of *fmincon*, each with a

different randomly specified initial condition. The parameter set associated with the smallest squared error was selected. We then used this parameter set to simulate behavior in Experiments 1 and 2 (Fig 4). The parameter sets for Models 1 and 2 are reported in Table 1.

We used bootstrapping to identify confidence intervals for each parameter in Models 1 and 2. To do this, we randomly sampled participants from Experiments 1 and 2, with replacement. For each random sample, we computed the mean behavior, and fit Models 1 and 2 to that mean behavior by minimizing Eq (9). We repeated this process 10,000 times, and selected the interior 95% of each parameter distribution. These intervals are also reported in Table 1.

The primary disagreement between Models 1 and 2 was their predictions regarding time-based decay after a set break. These predictions are quantified in Fig 4D. To quantify the magnitude of decay predicted by the model during acceleration and deceleration, we analyzed individual participant data. For each participant, we calculated the total set break decay predicted by Models 1 and 2, using the best fit parameters for that participant. For each participant, we calculated the predicted model decay after each set break, and then averaged across set breaks. In Fig 4D, we combined participants from Experiments 1 and 2 when quantifying set break decay during perturbation periods. However, only Experiment 1 was used in our analysis of error clamp period set-breaks because the error clamp period in Experiment 2 was during the period of spontaneous recovery.

We also statistically compared Models 1 and 2 via maximum likelihood estimation. We fit Model 1 and Model 2 to each participant's behavior in Experiments 1 and 2 by minimizing the cost function in Eq (8). To do this, we performed 100 iterations of *fmincon* in MATLAB R2018a (subject to the inequality constraints describing the conventional two-state dynamics), each time varying the initial parameter guess. We selected the parameter set across the 100 iterations that minimized squared error. For each participant, we used the model's RMSE and total trial count to calculate the log-likelihood for each fit. In S5A Fig, we compared the log-likelihood for Models 1 and 2. In addition, we also performed a model recovery analysis (S5B Fig). This analysis is detailed in the Supporting Information.

## Supporting information

**S1 Fig. Change in eye velocity during adaptation.** In the main text, we quantified adaptation as the perpendicular displacement of the eye during the acceleration and deceleration phases of the saccade. To calculate this displacement, we integrated horizontal velocity before and after peak speed. Here we show the horizontal velocity during various periods of the first block in Exp. 1. Data are within-subject change in velocity with respect to baseline. Vertical lines indicate timing of peak speed, thus separating the acceleration and deceleration phases of the movement. **A.** Horizontal velocity during the initial adaptation period. **B.** Effect of set break on horizontal and vertical velocities. The set break caused decay in horizontal velocity, primarily during the deceleration period (top). The decay was larger in the deceleration period ($-0.11 \pm 0.33$ deg/sec, $t(23) = -0.35$, $p = 0.73$ for the acceleration period, $-10.08 \pm 2.07$ deg/sec, $t(23) = -4.88$, $p = 6.33X10^{-5}$ for the deceleration period). Vertical velocity did not show decay following a set break (bottom, $+7.09 \pm 2.32$ deg/sec, $t(23) = 3.06$, $p = 0.006$). **C.** Re-learning after the set break during the first perturbation block. **D.** Decay of horizontal velocity during the first error clamp block. **E.** Timing of peak speed with respect to saccade onset. The timing that categorizes the movement into acceleration and the deceleration phases was computed by the peak speed, which depends on horizonal as well as vertical velocity. Thus, this timing could change over the learning period as the adaptation component increases, influencing the ratio of the two indexes between the acceleration and deceleration phases. We measured the timing of peak speed with respect to saccade onset in Exp. 1 as well as Exp. 2 and found that there

were no significant changes in peak speed timing (Exp 1: $F_{(2,69)} = 0.004$, $p = 0.997$; Exp 2: $F_{(3,156)} = 0.08$, $p = 0.97$). Combined data across the two experiments are plotted here. Shaded error regions are between subject SEM.
(TIF)

**S2 Fig. Acceleration period commands exhibited little or no decay during set-breaks.** Acceleration period commands generally adapted less than the deceleration period commands. This smaller magnitude could have made it more difficult to find significant changes due to set breaks. To check for this, we performed a series of analyses. **A**. Percent change in the perturbation blocks (Exp. 1 and 2), and error clamp blocks (Exp. 1). **B**. Absolute change in the perturbation blocks (Exp. 1 and 2), and error clamp blocks (Exp. 1). **C**. Here we controlled for differences in the adaptation extent exhibited by acceleration and deceleration period commands. We searched for a period in Exp. 1 where deceleration commands were most similar in magnitude to acceleration commands measured throughout the experiment. The optimal period occurred before the first error-camp period, where deceleration commands reached about 0.8 deg (pre-set break deceleration). Acceleration period commands reached about 0.75 deg throughout the experiment (pre-set break acceleration). Regardless, a decay was present in the deceleration period commands but not acceleration.
(TIF)

**S3 Fig. Comparison of adaptation due to horizontal errors and vertical errors in Experiment 2.** In Experiment 2, an initial perturbation block was followed by extinction and concluded with a block of error clamp trials. Half of the subjects experienced horizontal errors following vertical saccades (left panel), while the other half experienced vertical errors following horizontal saccades (right panel). **A**. Learning in response to horizontal (left panel) and vertical (right panel) errors. Subjects learned somewhat more from horizontal errors than from vertical, especially during the acceleration period (overall rate of change, $0.005 \pm 0.0004$ deg/trial vs. $0.002 \pm 0.0002$ deg/trial, within subject difference $t(39) = 7.32$, $p = 7.89 \times 10^{-9}$). **B**. Effect of set breaks during perturbation and error clamp periods, collapsed across all horizontal error subjects and vertical error subjects. Deceleration period commands decayed more than acceleration period commands during the perturbation period (within subject difference: $-0.22 \pm 0.07°$, $t(39) = -3.34$, $p = 0.001$), but neither exhibited decay during the error clamp period, as expected during spontaneous recovery.
(TIF)

**S4 Fig. Parameter recovery analysis.** Models 1 and 2 were fit to individual participants in Exp. 1 and Exp. 2, and then used in a parameter recovery analysis. This analysis was conducted via a bootstrapping approach with 1000 batches. In each batch, each parameter set ($n = 64$ total) was used to simulate noisy behavior with Model 1 (**A**) or Model 2 (**B**). Then Models 1 and 2 were fit to the simulated noisy data for each participant. The parameters recovered by each model were compared with the true underlying parameters. To compare the true parameters with the recovered parameters, we calculated Pearson's correlation coefficient across the 64 participants simulated in a given batch. We repeated this process 1000 times, varying the seed for the random number generator used in each noisy simulation. We then computed the median correlation coefficient, as well as 95% confidence intervals, for each parameter in Models 1 and 2. The higher this correlation, the better the fitting robustness. Black dots represent the median correlation coefficient across 1000 simulated batches, and lines represent 95% confidence intervals.
(TIF)

**S5 Fig. Statistical Model Comparison and Model Recovery Analysis. A.** We calculated the log-likelihood for observing each participant's data given Model 1 and Model 2. In panel **A** we subtract these two likelihoods for each participant: positive values indicate that Model 2 is more likely to explain the participant's data. **B.** We performed a model recovery analysis to determine if the log-likelihood was biased toward either Model 1 or 2, given the design of Experiments 1 and 2. We used a simulation to produce data either with Model 1 or 2, and then calculated the log-likelihood for the true model and the opposing model. For each simulation, we calculated the fraction of participants that BIC or AIC selected for a given model. We did this process 1000 times, each time re-simulating behavior for each subject. At left we simulated the experiments using Model 1. In the middle we simulated the experiments using Model 2. At right we show the probability that a participant was better described by Model 1 or Model 2 in the actual data. This model recovery analysis demonstrated that AIC and BIC were both dependable measures that could be used for model comparison; when data were simulated with Model 1, the log-likelihood measure was more likely to recover Model 1. When data were simulated with Model 2, the log-likelihood measure was more likely to recover Model 2. The bars show the mean. Lines indicated 95% confidence intervals across the 1000 simulations batches.
(TIF)

**S6 Fig. Accuracy of error clamp trials.** To test whether spontaneous recovery was due to differential decay of multiple learning processes and not to an imperfect implementation of error clamp trials, we calculated the average error in the direction of the original perturbation during the error clamp period of Experiment 2 for each subject and compared it to the average error in the same direction for that same subject during the baseline period (unperturbed trials). We found no difference in error between the two conditions (baseline error: $-0.006 \pm 0.07^0$ vs. error clamp: $-0.10 \pm 0.03^0$, within subject difference: $-0.10 \pm 0.08^0$, t(39) = $-1.24$, p = 0.22). It should be noted that the visual target occupied a square area of $0.25^0$, and these error measurements are with respect to the center of that area, so the gaze location is still within the bounds of the visual target under both conditions.
(TIF)

**S7 Fig. Change in horizontal starting position (CHSP) altered saccade displacement, mostly during acceleration.** The y-axis quantifies the trial-to-trial change in saccade trajectory during the acceleration and deceleration phases of the saccade. The x-axis shows CHSP of the saccade in trial n with respect to the saccade in the previous trial. **A.** Effect of change in horizontal starting position. Data from Exp. 3. Leftmost bin is CHSP < -0.5˚, next is -0.5˚ < CHSP < 0˚, third bin is 0˚ < CHSP < 0.5˚, and the rightmost bin is CHSP > 0.5˚. **B.** Close up of middle two bins in part A. **C.** Effect of change in starting position in the baseline periods of Experiments 1 & 2, when there are no errors present (similar to the mean zero perturbation of Exp. 3). Bins are quintiles of CHSP. As in Experiment 3, there is a strong effect that is mainly expressed during acceleration rather than deceleration. This observation adds further support to the idea that there may be separate controllers that contribute to acceleration and deceleration periods of a saccade.
(TIF)

## Author Contributions

**Conceptualization:** Simon P. Orozco, Reza Shadmehr.

**Data curation:** Simon P. Orozco.

**Formal analysis:** Simon P. Orozco, Scott T. Albert.

**Funding acquisition:** Reza Shadmehr.

**Validation:** Simon P. Orozco, Scott T. Albert.

**Visualization:** Scott T. Albert.

**Writing – original draft:** Simon P. Orozco, Scott T. Albert, Reza Shadmehr.

**Writing – review & editing:** Simon P. Orozco, Scott T. Albert, Reza Shadmehr.

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
