## [Decision Letter · Decision Letter 0]

7 Dec 2020

Dear Shadmehr,

Thank you very much for submitting your manuscript "Multiple timescales of motor memory during saccade adaptation" for consideration at PLOS Computational Biology.

As with all papers reviewed by the journal, your manuscript was reviewed by members of the editorial board and by several independent reviewers. In light of the reviews (below this email), we would like to invite the resubmission of a significantly-revised version that takes into account the reviewers' comments.

All reviewers highlight that this study is potentially of high interest. However, the reviewers highlight that a more thorough analysis of the data is desirable to form a solid basis for the modeling efforts. The interpretation of the modeling results with respect to the observations as well as based on model comparison also need clarification. I think these are addressable concerns and more information about these will be highly valuable for the community.

We cannot make any decision about publication until we have seen the revised manuscript and your response to the reviewers' comments. Your revised manuscript is also likely to be sent to reviewers for further evaluation.

Sincerely,

Gunnar Blohm, Ph.D.

Associate Editor

PLOS Computational Biology

Samuel Gershman

Deputy Editor

PLOS Computational Biology

Reviewer's Responses to Questions

**Comments to the Authors:**

Reviewer #1: This paper proposes a novel idea that even a single continuous movement might be generated from multiple controllers with different dynamic properties for adaptation. The data of experiment 3 clearly show that the first and the second halves of the adaptive system has different learning properties. The model-based analysis with the state-space representation of the memory update revealed that two adaptive system components are composed of the fast and the slow memories. Although the current version of the manuscript is written very well and the methods and the results are reasonable, there is a couple of points that should be addressed before it is accepted.

Major questions.

1) In general, how the hypothesis was derived should be written in the Introduction section. In the third paragraph, the authors stated that the timescale of learning was different among phases of a single movement by citing the previously published paper. However, it is unclear why the acceleration and the deceleration phases should separate the single movement and why separating the movement by time was not adopted. It could be arbitral and determined for the sake of simplicity in the analysis method. If this is the case, the author should examine how the analysis's main result is sensitive to how the phase was defined. The author might consider this separation based on a neurophysiological background. If this is the case, this background should be briefly stated in the Introduction.

Major 2. The timing that categorizes the movement by the acceleration and the deceleration phases was computed by the peak speed, sqrt( H’^2+V’^2). Thus, this timing should be changed over the learning period as the adaptation component H’ increases, which influences the learning the ration of the two indexes between the acceleration and deceleration phases. Apparently, in figure 1D, the peak of the H’ shifted from late (around 40ms) to early (around 20ms), which should influence peak speed timing. Thus the estimated relative contribution of the deceleration phase increases over the trial as the timing of the peak speed shifts. If this is the case, the different learning speed shown in figure 1E is the artifact of this timing shift. To refute this possibility, the authors should plot how the peak speed change over trials is timing.

Major 3. Model comparison based on BIC is solid. However, in general, relying on only one index is risky to lead the conclusion. As was done in many papers, reporting BF, AIC in addition to BIC is recommended.

Minor 1. I request additional references to the saccadic adaptation task in line 85.

Minor 2, L129. How long is the duration of the set-breaks?

Minor 3, L177. It took me a while to notice that (0,15) the target's position at (horizontal, vertical) in degrees. Just a brief note about this notation is helpful to read.

Minor 4, L717. The definition of the error is reasonable in terms of the state-space model of memory update. It is helpful to understand if the authors briefly describe how the brain observes the error in the saccadic adaptation task. Is it via a corrective saccade or the retinal slip of the target stimuli?

Minor 5, L800-L803. Why is using the individual commands rather than the total commands reasonable? How were these two costs examined before one was selected?

Minor 6, L849. How was the robustness of fitting examined?

Reviewer #2: This manuscript examines saccade adaptation from a behavioral and modeling perspective. Consistent with their prior findings, the authors note that during cross-axis adaptation, saccade kinematics change in a manner inconsistent with typical oblique saccades, having a much larger corrective component at the end of the movement that is more responsive to errors; additionally, both the early and late components exhibited spontaneous recovery. The authors noted that the popular two-state model is unable to capture these temporal dynamics, and instead propose an alternative model in which the early and late phases of the saccade are governed by separate controllers.

Unfortunately, there are 3 major issues in this manuscript. The first is that even under the best of circumstances (e.g., non-adapted saccades), the horizontal and vertical components of oblique saccades are not obligately coupled (e.g., King et al J Neurophysiol 1986, Becker and Jurgens Vis Res 1989, etc). Specifically, the two components of the saccade do not necessarily start and stop at the same time, their peak velocities do not align well, and the shorter component is typically stretched, possibly asymmetrically. Practically speaking, this poses a challenge when attempting to examine the acceleration and deceleration phases of an oblique saccade – particularly one generated via adaptation; it seems highly unlikely that the peak speed (considering the horizontal and vertical components together) is an appropriate place to divide the individual components into acceleration and deceleration phases. If a given component is not appropriately divided, this in turn could lead to misleading conclusions about the extent to which changes are occurring in the early and late phases of that component (e.g., if not enough of the true acceleration phase is labeled as such it could explain why changes are really only observed in the deceleration phase). This calls the main behavioral findings of this manuscript into question.

The second issue is that is already well known that the early and late phases of a saccade are critically distinct in how they are controlled: early in the movement the motor command primarily driven by a preplanned command, whereas late in the movement the motor command is influenced by corrections (e.g., based on a forward-model). Even if we ignored the issue above, differences in online corrections could potentially explain why different phases of the movement adapt to different extents; for example, it may have to do with whether adaptation drives changes in the forward model or the controller, rather than needing to invoke a model in which the early and late phases of a movement are driven by separate controllers. Indeed, the authors’ previous work regarding fatigue (e.g., Xu-Wilson 2008) suggests that the ability to generate late corrections is critical for maintaining saccades accuracy even in non-adapted saccades, and thus it is not surprising that the deceleration phase responds sooner and to a greater extent in response to observed errors. The authors thus need to consider the presence of late-movement motor corrections when attempting to explain their findings (e.g., lines 268-275).

The third issue is that the authors base much of their judgments of model accuracy on transient changes during set breaks. However, this leads to two potential problems; one is that the authors average set-break behavior for error clamps across experiments 1 and 2, but the expected decay over those set breaks is expected to be in the opposite direction (decay toward baseline in experiment 1, or decay toward the adapted state – spontaneous recovery – during experiment 2). Thus by averaging these two datasets together the authors are likely wiping out any potential decay that might be present (e.g., compare Fig. 4B to Fig. 2A). The other potential issue is that if indeed there is much less adaptation occurring in the acceleration phase, forgetting may be present but so small that it is not reliably detectable relative to the noise in the data (e.g., if forgetting is proportional to the amount of adaptation achieved); this could lead to the erroneous conclusion that there is only decay in the deceleration phase.

For the reasons above, it is unclear what exactly to make of the authors’ behavioral findings, as well as the modeling efforts based on those behavioral results. The additional clarity provided by addressing these concerns is necessary before one can fully evaluate the authors' claims and their modeling efforts.

Additional Comments:

line 124: Running a MANOVA across so many trial bins is difficult to interpret as there is no sense of exactly which bins are driving the main effect; thus to draw a conclusion about a systematic change over time seems a little inappropriate from such an analysis.

lines 151-153: It isn’t clear why these subjects should be excluded. The authors should report how many participants were removed in this manner. However, this method is problematic because it will bias the results toward finding significant differences.

line 343: Why did the authors choose to compare trial n and trial n+1, versus using the triplet method that others have used previously (e.g., comparing trial n-1 to trial n+1)? Trial n is potentially contaminated by the surprising target jump (which is fairly obvious in the cross-axis case).

line 372-373: Why analyze when the correction moved in the direction of the error, versus the time when the saccade trajectory differed from baseline? It isn’t entirely clear why a saccade might deviate in the wrong direction early in the movement, but it could be deliberate (e.g., to compensate for ocular dynamics/inertia). In particular, it is interesting to note that this initial motion in the wrong direction does not appear in the first trial of experiments 1 and 2 (fig. 6F), suggesting it may indeed be a response to the fluctuating single-trial condition.

line 389: Does Model 2 predict that movement in the acceleration phase should go in the opposite direction in Experiment 3? If not, it seems that neither model is doing a good job of explaining the observed behavior, and there remains no clear winner between the two proposed models.

section starting 428: This model is reminiscent of the old bang-bang controller models of the cerebellum (in which one side of the cerebellum accelerated the eye and the other side of the cerebellum decelerated it); the authors should review this literature and consider how their model is similar or different to those models. It would also be helpful if the authors could better explain in their model how the same circuit can govern both the acceleration and deceleration phases of a movement depending on the sign of the error and the direction of the primary saccade (e.g., based on lines 445-455, presumably the right fastigial nucleus responding to anti-preferred rightward errors and affecting acceleration would also respond to preferred leftward errors and affect deceleration – how would that work?).

Reviewer #3: The study investigates behavioural signatures of fast and slow processes during saccadic motor learning that go back to the occurrence of spontaneous recovery after extinction. I have reviewed this manuscript before for a different journal and I think that in the present version the authors much better present what their data can actually show. It is a fine-scale examination how fast and slow processes are expressed in the saccadic acceleration and the deceleration phase during learning, set breaks and error-clamp trials. The manuscript now builds up this aim in the introduction and focusses on this strength in the thorough analysis of experimental data. Yet, I have continuing concerns about the modelling part. From the modelling the authors aims to conclude that learning relies on two adaptive controllers for acceleration and deceleration that are each supported by a slow and a fast process. This is an interesting hypothesis. However, in my understanding, the analysis shows that this hypothesis is not confirmed — model 2 is not superior to the simpler model 1. Still, the authors argue otherwise. As this remains a central point about the conclusions are drawn from the study, I would like to ask for clarification on this point.

Model 1 is the simpler model in which a slow and a fast process are expressed to a different amount in acceleration and deceleration (2 * trial-based retention + 2 * error sensitivity + 2 * time-based retention + 1 * percentage of expression within acceleration/deceleration = 7 parameters). Model 2 should represent the alternative hypothesis - that acceleration and deceleration are each controlled by different fast and slow processes.

Firstly, it it important to note that the exact version of model 2 is generated based on a preceding comparison of different model variants in the methods section, examining different combinations of shared parameters across the four states. The final version consists of 2 * trial-based retention (fast and slow, shared across acceleration and deceleration) + 4 * error sensitivity (fast and slow for each acceleration and deceleration) + 2 * time-based retention (a fast deceleration parameter and a shared one for the remaining three combinations) = 8 parameters. This analysis is suitable but please note that the final version of model 2 reflects a complex interplay of different rates in which a clear separation between between acceleration and deceleration with each fast and slow is reduced to error sensitivity. Trial-based retention is shared between acceleration and deceleration and time-based retention is special for fast deceleration but the same for the other three states (slow acceleration, fast acceleration and slow deceleration).

The following comparison of model 2 with model 1 reveals that both models fit well to the data. This is strongly confirmed by visual inspection. The only small difference that the authors claim is separating goodness of fit between model 1 and 2 lies in set-breaks during learning (not during learning itself, nor during error-clamp or set-breaks during error-clamp). Hence, this small difference relies on temporal decay — a parameter of model 2 that is not specific to the hypothesis.

The authors put forward that BIC analysis results are mixed but that the mean over subjects of BIC difference between model 1 and 2 favors model 2, supported by a t-test. Taking BIC differences is rather unusual and Fig. S2A clearly shows that even then, the distribution is highly skewed. As the authors admit, model 1 was superior for 35 subjects and model 2 for 29 subjects, also shown in Fig. S2B on the right side (model recovery analysis based on the data, 55% to 45% for model 1). If I understand it correctly, the model recovery analysis reveals once more that BIC favors model 1 over model 2 if a) fitted to the real data, b) if fitted to simulated data based on model 1 and c) if fitted to simulated data based on model 2 — simply because model 2 is less complex (one parameter less). Isn’t this exactly what BIC should provide, i.e. correction for model complexity? I cannot understand why the authors argue that the analysis reveals a bias of BIC such that it would not be appropriate in means of model comparison (“BIC becomes biased towards the model with fewer parameters.”) This is not a bias but a benefit of BIC and the reason to use it.

Experiment 3 is then presented as a further support for the idea of distinct controllers for acceleration and deceleration as supported by model 2. Experimentally, it reveals a bimodal response in which the acceleration phase of the saccade changes in the opposite direction of error while the deceleration phase changes in the direction of error. This result is interesting, indeed. However, to my mind it cannot clarify the lack of evidence from the modelling. The effect in experiment 3 is in learning from error (not in time-based decay) and specifically learning from single error which is shown to be different from learning of persistent errors (experiment 1-2 with modelling).

Please correct me if I got anything wrong. I put forward these points because I think we should really be careful with the conclusions that we draw from modelling. I find the data of the present experiment worthwhile to report and I also think that the separate controller hypothesis is interesting and worth testing. If the outcome does not provide evidence in favour of it this should simply be stated accurately.

**Have all data underlying the figures and results presented in the manuscript been provided?**

Reviewer #1: Yes

Reviewer #2: Yes

Reviewer #3: Yes

PLOS authors have the option to publish the peer review history of their article (what does this mean?). If published, this will include your full peer review and any attached files.

Reviewer #1: No

Reviewer #2: No

Reviewer #3: No
---

## [Decision Letter · Decision Letter 1]

7 Mar 2021

Dear Shadmehr,

Thank you very much for submitting your manuscript "Spontaneous recovery of motor memory during saccade adaptation" for consideration at PLOS Computational Biology.

As with all papers reviewed by the journal, your manuscript was reviewed by members of the editorial board and by several independent reviewers. In light of the reviews (below this email), we would like to invite the resubmission of a significantly-revised version that takes into account the reviewers' comments. 

In partiular, I would like to invite the authors to address the remaining concerns raised by reviewers 2 and 3 in a constructive fashion. Potential alternative interpretations and limitationsof the study as raised should be clearly and explicitly discussed. Given that, I would also like to invite the authors to reflect on whether the study title and abstract could be more nuanced. 

We cannot make any decision about publication until we have seen the revised manuscript and your response to the reviewers' comments. Your revised manuscript is also likely to be sent to reviewers for further evaluation.

Sincerely,

Gunnar Blohm, Ph.D.

Associate Editor

PLOS Computational Biology

Samuel Gershman

Deputy Editor

PLOS Computational Biology

Reviewer's Responses to Questions

**Comments to the Authors:**

Reviewer #1: The authors revised the paper following my suggestion enough. I recommend this paper to get accepted.

Reviewer #2: The authors have answered the majority of my concerns in great detail and in a satisfactory manner. However, a two concerns remain.

First, the authors noted in their response to point 11 that there is an untested possibility related to my initial concern, largely that the same behavior could have arisen by a simpler alternative model in which the controller and the forward model partially adapt (particularly if they adapt to different extents). In theory this could result in different degrees of expressed adaptation during the acceleration and deceleration phases, since the controller primarily dominates the response during the acceleration phase while the forward model comes online to steer corrections later in the movement. If the two systems partially adapt to different extents, and if one exhibits temporal decay while the other only exhibits decay in response to trials (e.g., the work of Smith and colleagues), this could arguably explain the results without having to resort to a more complex model with separate two-state controllers for the acceleration and deceleration phases. It would be helpful if the authors considered this simpler alternative. Indeed, the authors seem to discourage this possibility by arguing that the superior colliculus doesn’t show changes in activity; however they then contradict themselves by proposing that the colliulus could be involved in remapping (point 18). As an aside, it is also important to note that the colliulus represents saccades in 2D but saccade motor commands are subsequently modified (e.g., to reflect the torsion component to obey Listing's law) before the commands are sent to the burst generators; thus there is yet another stage after the superior colliculus at which the motor command may differ between adapted and non-adapted saccades.

The other potential area of concern is that the authors primarily depend on interpretation of a null finding (i.e., lack of a significant change in the acceleration command) to disentangle their models. I would caution the authors about over-emphasizing the lack of observing a significant change as being evidence in favor of no change, and particularly in relying on this finding as the primary means of differentiating between models. It would be helpful if the authors consider how they phrase their findings and their model comparisons in light of this concern, and they should acknowledge in the manuscript that they are relying on interpretation of a null finding as evidence in favor of a lack of a change, and in light of that need to be cautious of the conclusions drawn.

Reviewer #3: I thank the authors for their detailed responses and improvement of the manuscript. In response to the criticism that the BIC analysis argued for model 1 instead of model 2, the authors added two learning parameters for error sensitivity to both models. Hence, error sensitivity can increase over time. Firstly, it improved the fit of both models to the data and, secondly, it shifted model superiority according to BIC from model 1 to model 2 (BIC: 37 vs. 27 subjects in favor of model 2; AIC: 45 vs. 19 subjects in favor of model 2). I much appreciate this improvement, the effort taken for the manuscript revision, and the contribution that the data can provide, i.e. a fine-scale examination of how fast and slow processes are expressed throughout the acceleration and deceleration phase of the saccade. However, I must admit that the authors could not rule out my concern about the validity of model 2 with respect to the storyline of the manuscript.

The story that the authors make is that learning and spontaneous recovery rely on two adaptive controllers, one for acceleration and one for deceleration, that each rely on a fast and a slow process. Hence, they suggest a four-state framework as an extension of the well-known two-state framework of a fast and a slow state of learning. My doubts in the validity of model 2 for the four-state framework relied on two aspects that still persist:

1) With the aim of parameter reduction, model 2 is generated based on a preceding comparison of different model variants. In the revised version, the final version of model 2 is still a complex interplay of different rates that are shared across acceleration and deceleration or fast and slow states in the following form: 2 trial-based retention rates (fast and slow; hence shared between acceleration and deceleration), 4 error sensitivity parameters (the only parameter with differentiation for all 4 states), 2 parameters for error sensitivity increase (one for acceleration, one for deceleration; hence shared between fast and slow) and 2 time-based retention rates (fast deceleration and slow deceleration; fast and slow acceleration are fixed to 1 such that they do not decay and do not need a parameter). Thus, learning and decay of fast and slow processes spread in a complex way across different phases of the saccade. However, I cannot conclude a clear separation between an acceleration and a deceleration controller, each relying on a fast and a slow state. Firstly, because the parameter reduction of model 2 does not support the hypothesis of a clear four-state framework, and secondly, because the separation of the saccade into the acceleration and deceleration phases has not been tested against other separations of the saccade. Thirdly, parameter reduction across fast and slow states was not tested for model 1.

2) The effect that separates model 1 and 2 (that both fit well according to visual inspection) relies on set-breaks only, and hence, exclusively on time-based decay in the models. This parameter is not differentiated across all 4 states in model 2, and is hence not specific to the hypothesis. Moreover, the effect is not present during spontaneous recovery. Of course, a separation into fast and slow states is a current and well-known explanation of spontaneous recovery. However, the additional differentiation that the authors make concerning the acceleration and deceleration of the saccade refers to time-based decay during set-breaks, not the phenomenon of spontaneous recovery. I fully support the result that during set-breaks, decay is more present in the deceleration than in the acceleration phase of the saccade, as shown by the experimental data and the models. However, from this interesting detail about temporal decay when no saccades are made, I cannot conclude a general four-state framework of learning and spontaneous recovery. Moreover, I cannot support that in the revised version, spontaneous recovery appears in the title and as a focus in the introduction.

I do not want to stand in the way of publishing a study that took so much effort, and I really appreciate the work that was done for this study. But I feel that these issues at least need to be discussed and clearly acknowledged in the manuscript if the authors want to insist on their storyline.

**Have all data underlying the figures and results presented in the manuscript been provided?**

Reviewer #1: Yes

Reviewer #2: Yes

Reviewer #3: Yes

PLOS authors have the option to publish the peer review history of their article (what does this mean?). If published, this will include your full peer review and any attached files.

Reviewer #1: No

Reviewer #2: No

Reviewer #3: No
---

## [Editor Report · Decision Letter 2]

13 Jun 2021

Dear Shadmehr,

We are pleased to inform you that your manuscript 'Adaptive control of movement deceleration during saccades' has been provisionally accepted for publication in PLOS Computational Biology.

Best regards,

Gunnar Blohm, Ph.D.

Associate Editor

PLOS Computational Biology

Samuel Gershman

Deputy Editor

PLOS Computational Biology

---

## [Editor Report · Acceptance letter]

30 Jun 2021

PCOMPBIOL-D-20-02039R2 

Adaptive control of movement deceleration during saccades

Dear Dr Shadmehr,

I am pleased to inform you that your manuscript has been formally accepted for publication in PLOS Computational Biology. Your manuscript is now with our production department and you will be notified of the publication date in due course.

With kind regards,

Katalin Szabo
